# CLUH controls astrin-1 expression to couple mitochondrial metabolism to cell cycle progression

Désirée Schatton[1,2], Giada Di Pietro[1,2], Karolina Szczepanowska[2,3†], Matteo Veronese[1,2,4], Marie-Charlotte Marx[1,2], Kristina Braunöhler[1,2], Esther Barth[1,2], Stefan Müller[2], Patrick Giavalisco[4], Thomas Langer[2,5], Aleksandra Trifunovic[2,3,4], Elena I Rugarli[1,2,4]*

[1]Institute for Genetics, University of Cologne, Cologne, Germany; [2]Cologne Excellence Cluster on Cellular Stress Responses in Aging-Associated Diseases, University of Cologne, Cologne, Germany; [3]Institute for Mitochondrial Diseases and Ageing, Medical Faculty, University of Cologne, Cologne, Germany; [4]Center for Molecular Medicine, University of Cologne, Cologne, Germany; [5]Max Planck Institute for Biology of Ageing, Cologne, Germany

*For correspondence:
elena.rugarli@uni-koeln.de

Present address: †ReMedy International Research Agenda Unit, IMol Polish Academy of Sciences, Warsaw, Poland

**Competing interest:** The authors declare that no competing interests exist.

**Abstract** Proliferating cells undergo metabolic changes in synchrony with cell cycle progression and cell division. Mitochondria provide fuel, metabolites, and ATP during different phases of the cell cycle, however it is not completely understood how mitochondrial function and the cell cycle are coordinated. CLUH (clustered mitochondria homolog) is a post-transcriptional regulator of mRNAs encoding mitochondrial proteins involved in oxidative phosphorylation and several metabolic pathways. Here, we show a role of CLUH in regulating the expression of astrin, which is involved in metaphase to anaphase progression, centrosome integrity, and mTORC1 inhibition. We find that CLUH binds both the *SPAG5* mRNA and its product astrin, and controls the synthesis and the stability of the full-length astrin-1 isoform. We show that CLUH interacts with astrin-1 specifically during interphase. Astrin-depleted cells show mTORC1 hyperactivation and enhanced anabolism. On the other hand, cells lacking CLUH show decreased astrin levels and increased mTORC1 signaling, but cannot sustain anaplerotic and anabolic pathways. In absence of CLUH, cells fail to grow during G1, and progress faster through the cell cycle, indicating dysregulated matching of growth, metabolism, and cell cycling. Our data reveal a role of CLUH in coupling growth signaling pathways and mitochondrial metabolism with cell cycle progression.

## Editor's evaluation

The work is of interest to cell biologists studying metabolism and its regulation during the cell cycle. It reveals how CLUH, a protein involved in mitochondrial function regulation and metabolism, regulates the levels of astrin, a protein functionally involved in cell division and mTOR regulator, integrating metabolism and cell cycle.

## Introduction

Proliferating cells adjust their metabolism to each individual phase of the cell cycle (*DeBerardinis et al., 2008*; *Buchakjian and Kornbluth, 2010*; *Lee and Finkel, 2013*; *Salazar-Roa and Malumbres, 2017*). Before division cells must double their biomass and DNA content. This entails a prominent rewiring of metabolism to foster biosynthetic pathways allowing the synthesis of nucleic acids,

proteins, and lipids. During mitosis, an increase in ATP production supplies the energy needed for pulling apart and segregating the chromosomes. As metabolic organelles, mitochondria are key actors in the metabolic reprogramming of proliferating cells. On the one hand, cell cycle regulators influence mitochondrial metabolism in various ways (*Leal-Esteban and Fajas, 2020*). For example, CDK1 phosphorylates components of the respiratory chain during G2 and of the mitochondrial translocase of the outer membrane during mitosis, probably to enhance oxidative respiration and ATP production (*Harbauer et al., 2014*; *Wang et al., 2014*). On the other hand, energy deficiency is sensed by cell cycle checkpoints causing cell cycle arrest (*Mandal et al., 2005*; *Salazar-Roa and Malumbres, 2017*). Mitochondrial dynamics is also intimately linked to cell cycle progression. A hyperfused mitochondrial network is important for the G1–S transition, while mitochondria must fragment before cells enter mitosis (*Mitra et al., 2009*). Therefore, proliferating cells depend on several coordinated mechanisms to regulate how mitochondria utilize nutrients to fuel cell growth or energy production during the cell cycle. However, how mitochondrial metabolism and respiratory function are coordinated during the cell cycle is still largely unclear. Unraveling these mechanisms is crucial since they are often dysregulated in cancer.

One of the ways, how mitochondrial metabolism can be rewired is through regulation of RNA regulons by RNA-binding proteins (*Schatton and Rugarli, 2018*). CLUH (clustered mitochondria homolog) is a conserved RNA-binding protein, which binds transcripts encoding proteins involved in the respiratory chain, the tricarboxylic acid (TCA) cycle, and other mitochondrial metabolic pathways (*Gao et al., 2014*). Upon loss of CLUH, target mRNAs are subjected to faster decay and their respective proteins are decreased in abundance (*Schatton et al., 2017*). This leads to alterations in mitochondrial distribution (mitochondrial clustering as the gene name alludes to) mitochondrial cristae integrity, respiratory defects, loss of mtDNA, and decreased activity of TCA cycle enzymes (*Schatton et al., 2017*; *Wakim et al., 2017*; *Pla-Martín et al., 2020*). In the mouse, constitutive loss of *Cluh* causes neonatal lethality, while a liver-specific *Cluh* knockout (KO) led to hypoglycemia and defective ketogenesis upon starvation (*Schatton et al., 2017*). CLUH and its target mRNAs form ribonucleoprotein particles in primary hepatocytes. These CLUH granules are important not only to protect target mRNAs from degradation, but also to restrict mTORC1 activation and promote mitophagy (*Pla-Martín et al., 2020*). Thus, in postmitotic hepatocytes CLUH controls the balance between catabolism and anabolism, allowing a survival response to starvation.

Astrin (also called SPAG5) is a multifunctional protein with a well-established role during mitosis. The complex astrin/kinastrin/DYNLL1 is involved in the stabilization of the kinetochore–microtubule interactions in metaphase (*Kern et al., 2017*). Consistently, cells lacking astrin or its interactor kinastrin show defects in chromosome segregations and a delayed metaphase to anaphase progression (*Gruber et al., 2002*; *Dunsch et al., 2011*). Astrin has however other roles in interphase, ranging from mTORC1 inhibition (*Thedieck et al., 2013*), centrosome integrity and centriole duplication (*Thein et al., 2007*; *Kodani et al., 2015*), and regulation of p53 levels during G2 after DNA damage (*Halim et al., 2013*). Astrin is highly expressed in several cancers and it is considered a poor prognostic marker (*Yuan et al., 2014*; *Abdel-Fatah et al., 2016*; *Bertucci et al., 2016*; *Zhou et al., 2018*; *Ying et al., 2020*).

We show here that in HeLa cells astrin exists in two isoforms, astrin-1 and astrin-2. We reveal that CLUH binds the *SPAG5* mRNA, as well as the astrin-1 protein. The interaction of CLUH and astrin-1 occurs during interphase and protects astrin-1 and kinastrin from degradation. Moreover, we show that CLUH positively regulates mitochondrial anaplerotic pathways that sustain mTORC1 activation. Our data disclose a role of CLUH in coupling mitochondrial metabolism to cell cycle progression.

## Results
### CLUH interacts with astrin-1 and kinastrin
We previously showed that CLUH binds several mRNAs for mitochondrial proteins (*Gao et al., 2014*). However, among CLUH-bound transcripts, we found *SPAG5* encoding astrin, a multifunctional protein involved in metaphase to anaphase progression, centrosome integrity, and mTORC1 regulation (*Gruber et al., 2002*; *Thein et al., 2007*; *Schmidt et al., 2010*; *Dunsch et al., 2011*; *Thedieck et al., 2013*; *Gao et al., 2014*; *Kodani et al., 2015*). Surprisingly, we also detected astrin as an RNA-independent interactor of endogenous CLUH in a pull-down experiment in HeLa cells after

stable isotope labeling in cell culture (SILAC) followed by mass spectrometry (*Supplementary file 1*). Reciprocal immunoprecipitations (IPs) using antibodies against human CLUH and astrin, followed by western blot, confirmed the interaction (*Figure 1A*). Binding was also validated between CLUH and kinastrin (also known as SKAP), a known astrin interactor (*Figure 1B*; *Schmidt et al., 2010*; *Dunsch et al., 2011*).

Astrin appears as two isoforms (hereafter referred to as astrin-1 and astrin-2) in western blots, and CLUH interacts preferentially with the slower migrating isoform, astrin-1 (*Figure 1A*). The existence of these two isoforms has been reported in the literature but their identity remains enigmatic. On closer inspection of the *SPAG5* mRNA, we noticed the presence of five in-frame ATGs downstream of the first ATG (hereafter referred to as ATG2-6), which could serve as alternative translation start codons (with moderate and good Kozak sequences) (*Figure 1C*). Moreover, an upstream ORF within the *SPAG5* 5′ UTR (*Figure 1C*) could influence translation from the downstream ATGs as reported for other genes (*Morris and Geballe, 2000*; *Young and Wek, 2016*). To test these hypotheses, we produced different constructs expressing the 5′ UTR and the coding region of the *SPAG5* cDNA, and we mutagenized the uATG, the ATG1, or downstream ATGs individually or in different combinations to GGG (*Figure 1—figure supplement 1A*). When we expressed the uATG$^{GGG}$-SPAG5 construct, the expression of astrin-2 was suppressed (*Figure 1D*). Furthermore, mutagenesis of ATG1 confirmed the existence of a downstream alternative translation start site (*Figure 1D*). Overexpression of the constructs harboring mutagenized downstream ATGs revealed that astrin-2 is the product of alternative translation starting from ATG3-5 (*Figure 1E*). The identity of astrin-1 was also confirmed by pull-down of this isoform only using an antibody directed against the N-terminal region of the protein (*Figure 1—figure supplement 1B*).

To evaluate whether the interaction domain of astrin that binds to CLUH lies in the N-terminal part of astrin-1, we transfected a construct expressing both astrin-1 and astrin-2 (ATG1-SPAG5, *Figure 1—figure supplement 1A*) or an N-terminus deleted variant (Δ151-SPAG5, *Figure 1—figure supplement 1A*) and performed reciprocal IPs in HeLa wildtype and *CLUH* KO cells (*Wakim et al., 2017*; *Figure 1F*). An interaction was observed only with full-length astrin, proving that CLUH binds within the N-terminus of astrin-1 (*Figure 1F*). In contrast, the astrin interactor kinastrin has no binding preference to any of the two astrin isoforms, consistent with the previously mapped interaction domain between these two proteins (residues 482–693 of astrin) (*Figure 1—figure supplement 1C*; *Dunsch et al., 2011*; *Friese et al., 2016*; *Kern et al., 2016*; *Kern et al., 2017*). CLUH still interacts with astrin-1 upon kinastrin depletion, suggesting that kinastrin is not required for the interaction (*Figure 1—figure supplement 1D*). Overexpression of both astrin-1 and astrin-2 led to a diffuse cytosolic staining and to the formation of granular structures that vary in size, likely depending on the level of expression. Endogenous CLUH was characteristically recruited to astrin-1 but not astrin-2-positive structures, in agreement with the biochemical characterization of this interaction (*Figure 1G*).

In polysome profiles performed after chemical crosslinking, CLUH migrated in lighter fractions, as well as in fractions containing the monosome and polysomes, while astrin and kinastrin were only detected in the lighter fractions (*Figure 1H*), indicating the existence of distinct CLUH complexes. In conclusion, we identify the expression of two astrin isoforms via alternative initiation of translation, and establish CLUH as a specific molecular partner of astrin-1.

## CLUH controls astrin-1 and kinastrin stability and astrin expression

In absence of CLUH, kinastrin and astrin-1, but not astrin-2, are decreased in abundance (*Figure 2A*). To investigate whether CLUH influences the stability of its interaction partners, we performed a cycloheximide (CHX) chase in wildtype and *CLUH* KO HeLa cells (*Figure 2A*). In absence of CLUH, kinastrin and astrin-1, but not astrin-2, are highly unstable (*Figure 2A–D*). The instability of astrin in absence of CLUH was confirmed in *Cluh* KO MEFs which only express the larger isoform of astrin (*Figure 2—figure supplement 1A, B*). Inhibition of the proteasome by MG132 treatment in *CLUH* KO HeLa cells fully rescued kinastrin and partially astrin-1 levels (*Figure 2A, E–G*). Decreased kinastrin levels upon astrin depletion have already been reported (*Schmidt et al., 2010*; *Dunsch et al., 2011*).

The underlying reason for the only partial rescue of astrin-1 might be another level of regulation by CLUH. Given the identification of *SPAG5* mRNA as a CLUH target (*Gao et al., 2014*), we tested if CLUH is involved in astrin synthesis. To this end, we immunoprecipitated newly synthesized astrin in wildtype and *CLUH* KO cells after labeling with $^{35}$S-methionine for different time points (*Figure 2H–I*).

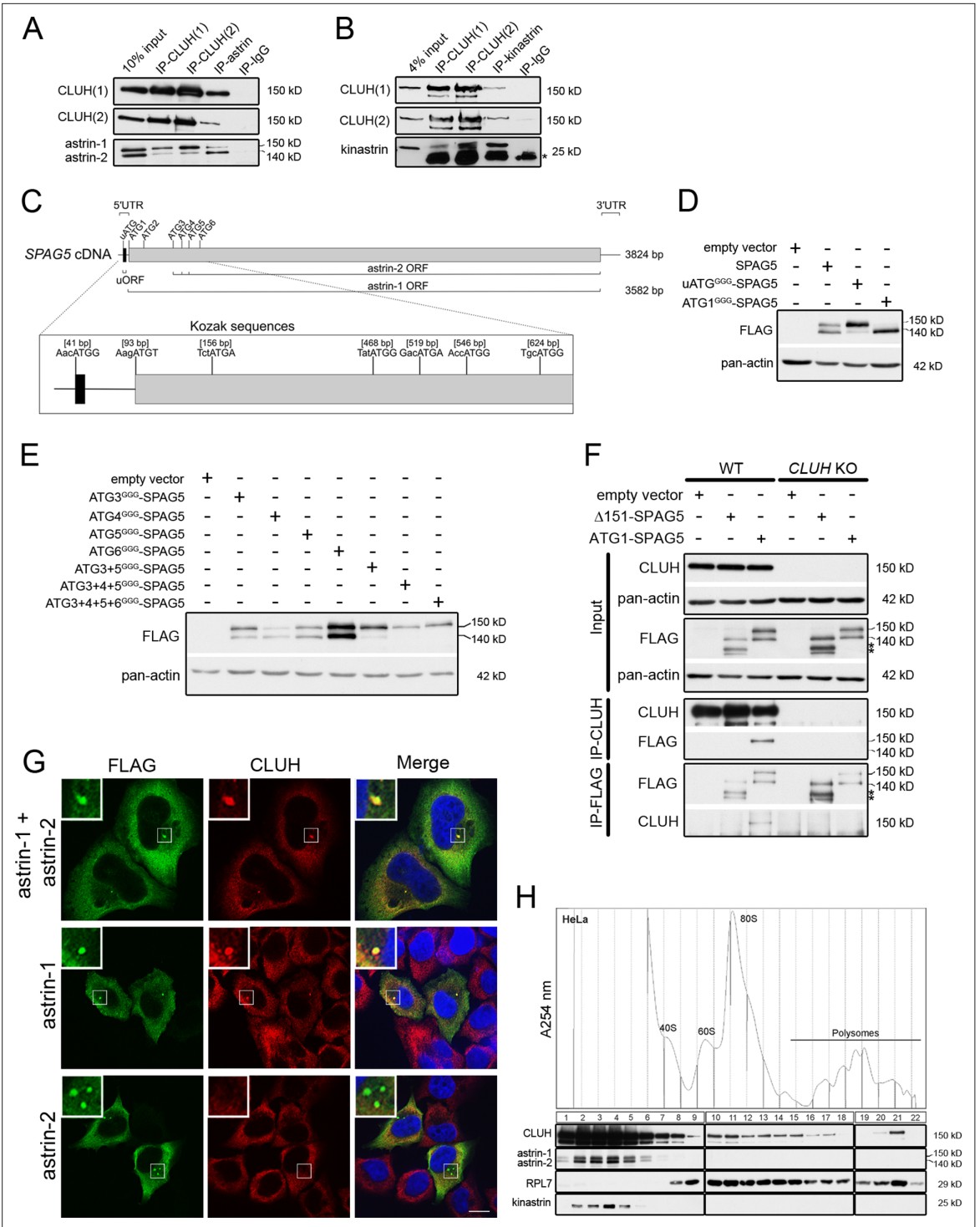

**Figure 1.** CLUH interacts with astrin-1 and kinastrin. (**A, B**) Western blots of reciprocal co-IPs of endogenous CLUH, astrin, and kinastrin in HeLa cells. Two different antibodies have been used to pull down CLUH (1, 2). Asterisk marks IgG light chain. (**C**) Scheme of human *SPAG5* cDNA with indicated UTRs and ORFs. Closeup shows positions of the uATG and ATG1–6 with surrounding Kozak sequences. (**D, E**) Western blots of HeLa cells overexpressing FLAG-tagged astrin constructs. Pan-actin was used as loading control. (**F**) Western blots of reciprocal co-IPs of endogenous CLUH and overexpressed FLAG-tagged astrin full length (ATG1-SPAG5) or a N-terminal deleted variant (Δ151-SPAG5) in WT and *CLUH* KO HeLa cells. Pan-actin was used as loading control for input samples. Asterisks indicate additional astrin bands appearing upon overexpression of the N-terminal deleted variant. (**G**) Confocal immunofluorescence pictures of HeLa cells overexpressing FLAG-tagged astrin-1 and astrin-2 (SPAG5), astrin-1 (ATG3 + 4 + 5<sup>GGG</sup>-SPAG5) or astrin-2 (ATG1<sup>GGG</sup>-SPAG5) alone stained for FLAG (green) and CLUH (red). DAPI was used to stain nuclei (blue). Small boxes in left corners show a ×4 magnified area of boxed regions. Scale bar, 10 μm. (**H**) Polysome profiling of HeLa cells chemically crosslinked with dithiobis succinimidyl

*Figure 1 continued on next page*

*Figure 1 continued*

propionate (DSP). At the top, absorbance profile at 254 nm of the fractions is shown with indicated peaks of 40S and 60S ribosomal subunits, 80S monosome and polysomes; at the bottom the corresponding western blots of the fractions are shown. RPL7 was used as a marker for ribosomes.

The online version of this article includes the following source data and figure supplement(s) for figure 1:

**Source data 1.** Uncropped blots for *Figure 1A,B,D,E, F*.

**Source data 2.** Uncropped blots for *Figure 1H*.

**Source data 3.** Unedited blots for *Figure 1A,B,D,E, F,H*.

**Figure supplement 1.** CLUH interacts with full-length astrin independently of kinastrin.

**Figure supplement 1—source data 1.** Uncropped blots for *Figure 1—figure supplement 1B–D*.

**Figure supplement 1—source data 2.** Unedited blots for *Figure 1—figure supplement 1B–D*.

We observed that newly synthesized astrin-1 is reduced in absence of CLUH to a greater extent than astrin-2, possibly also reflecting the decreased stability of this isoform (*Figure 2H–I*). CLUH controls the expression of some of its target mRNAs by preventing their degradation (*Schatton et al., 2017*). To elucidate whether this is also the case for *SPAG5*, we labeled newly synthesized RNA with the uridine analog ethynyl uridine (EU) and chased it for 8 hr. We precipitated the EU-RNA with biotin-azide using Click-iT chemistry and quantified *SPAG5* mRNA levels with qRT-PCR. *SPAG5* mRNA was less stable in absence of CLUH, whereas steady-state levels and EU incorporation were unaffected (*Figure 2J–L*). In conclusion, our data reveal that CLUH controls the expression of astrin-1 at multiple levels, by binding and stabilizing its mRNA, by affecting its synthesis, and maintaining its stability.

## CLUH binds astrin-1 in interphase

Astrin controls cell cycle progression at various stages, raising the question if the complex CLUH–astrin-1 is cell cycle dependent. To this end, we immunoprecipitated CLUH from cells enriched in different phases of the cell cycle (*Figure 3A*, *Figure 3—figure supplement 1*). The effective synchronization of the cells was confirmed by analyzing the input levels of cyclin D3 (which increases in G1 and G2), pH3-Ser10 (M phase marker), and pRPS6-Ser235/236 (inhibited upon starvation) (*Figure 3A*). Astrin-1, but not astrin-2, was more abundant in S and G2, while CLUH levels were similar in the different phases of the cell cycle (*Figure 3A*). Both astrin isoforms are phosphorylated at multiple sites, leading to slower migration in sodium dodecyl sulfate polyacrylamide gel electrophoresis (SDS–PAGE) in prometaphase (PM) (*Figure 3A*; *Chang et al., 2001*; *Cheng et al., 2008*; *Chiu et al., 2014*; *Chung et al., 2016*; *Geraghty et al., 2021*). The CLUH/astrin-1 complex is enriched in S and G2 phases but not detected in PM, indicating that CLUH preferentially interacts with unmodified astrin-1 in interphase and that the interaction might be controlled by the phosphorylation status of astrin (*Figure 3A*).

To fully capture the CLUH interactome, we performed four biological independent IPs of CLUH in G2-synchronized wildtype cells followed by label-free mass spectrometry (*Figure 3B*, *Figure 3—figure supplement 1A*). The correct synchronization was confirmed by propidium iodide (PI) staining followed by flow cytometry (*Figure 3—figure supplement 1B, C*). This experiment confirmed astrin-1 and kinastrin as highly enriched interactors of CLUH (*Figure 3B*, *Supplementary file 2*). In addition, proteins previously shown to be part of the astrin/kinastrin complex, such as MYCBP, or linking the complex to the microtubule cytoskeleton were also detected in the precipitate. Among these proteins are CLASP1 and 2, two microtubule plus-end binding proteins, and DYNLL1, a retrograde motor protein (*Manning et al., 2010*; *Schmidt et al., 2010*; *Dunsch et al., 2011*; *Kern et al., 2016*; *Kern et al., 2017*; *Figure 3B*). Interestingly, among top CLUH interactors, we found the centrosomal proteins CEP170B and CEP44, suggesting a link of CLUH with the centrosome and the microtubules (*Figure 3B, C*).

Since astrin has been previously found at the centrosome and pericentriolar material (*Cheng et al., 2007*; *Thein et al., 2007*; *Kodani et al., 2015*), we explored if astrin-1 resides in these organelles, using respective markers, such as CEP43 and PCM1. To avoid analyzing cells with large astrin-1 aggregates, we focused on cells expressing lower levels of astrin-1. In some cells, astrin-1 labeled a dotty structure that was positive at one extremity for CEP43 and partially colocalized with PCM1 (*Figure 3D*). However, in other cells astrin-1-positive punctae were not decorated by centrosomal markers (*Figure 3D*). This suggests that overexpressed astrin-1 can accumulate at the centrosome,

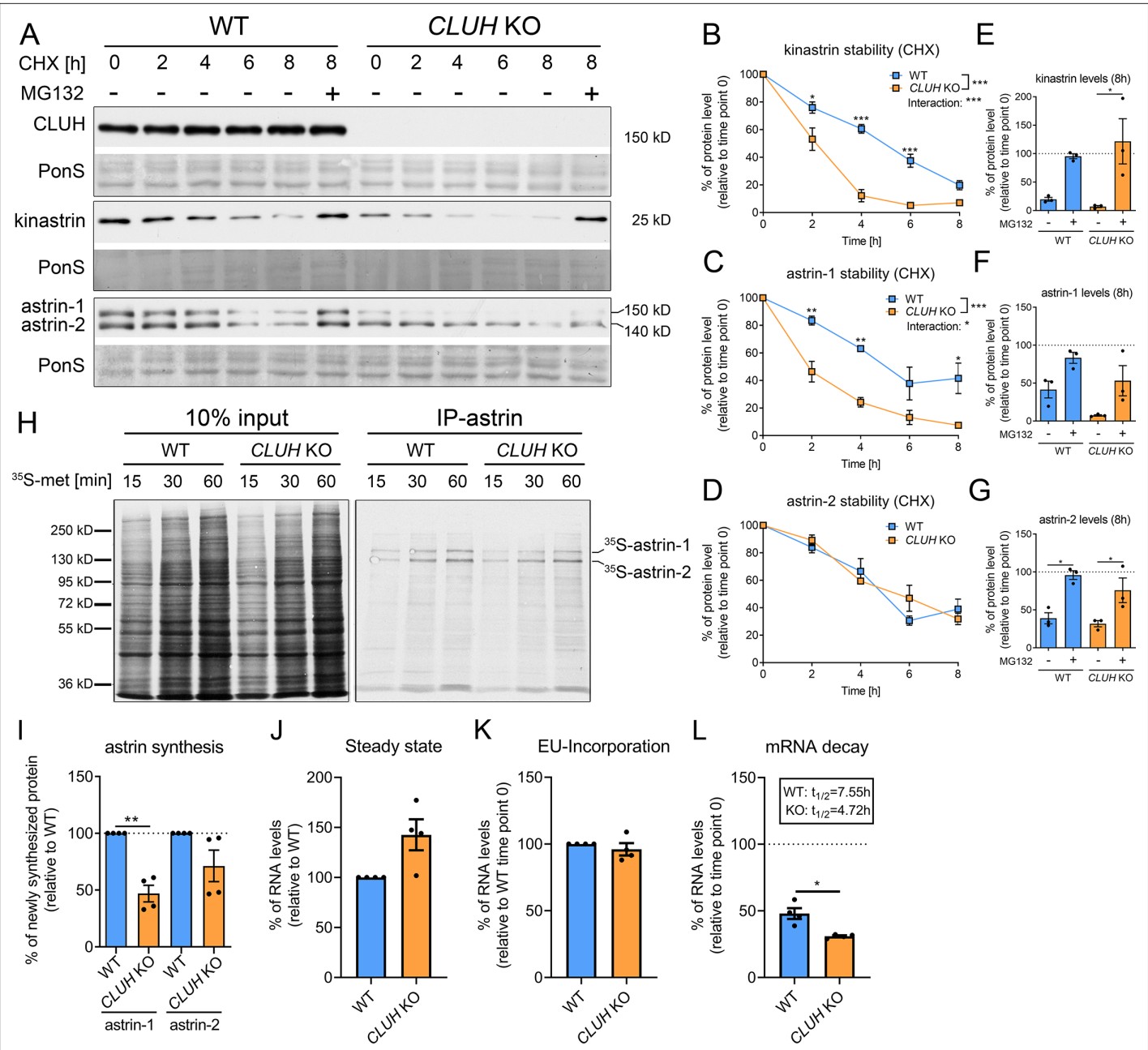

**Figure 2.** CLUH controls astrin-1 and kinastrin stability and astrin expression. (**A**) Western blots of WT and *CLUH* KO HeLa cells treated with CHX for indicated time points with or w/o additional MG132 treatment. Ponceau S staining was used as loading control. (**B–D**) Quantification of CHX chase western blots as shown in A ($n = 3$ independent experiments). Two-way analysis of variance (ANOVA) with post hoc Tukey's multiple comparison tests were performed with *$p \leq 0.05$; **$p \leq 0.01$; ***$p \leq 0.001$. Genotype × time interaction significance is also shown. Error bars represent standard error of the mean (SEM). (**E–G**) Quantification of proteins levels after 8 hr CHX treatment of western blots as shown in A ($n = 3$ independent experiments). Bars show the mean ± SEM. Dotted lines indicate protein levels at 0 hr time points. One-way ANOVA with post hoc Tukey's multiple comparison tests were performed with *$p \leq 0.05$. (**H**) Autoradiograms of IP of newly synthesized astrin labeled with $^{35}S$-methionine for indicated time points in WT and *CLUH* KO HeLa cells. (**I**) Quantification of immunoprecipitated newly synthesized astrin after 60 min labeling of experiments as shown in H ($n = 4$ independent experiments). Bars show the mean ± SEM. Two-tailed paired Student's *t*-test was performed with **$p \leq 0.01$. (**J**) Steady-state mRNA levels of *SPAG5* in WT and *CLUH* KO HeLa cells ($n = 4$ independent experiments). Bars show mean ± SEM. *RPL13* levels have been used for normalization. Levels of EU-incorporation (**K**) and mRNA decay (**L**) of *SPAG5* mRNA in WT and *CLUH* KO HeLa cells after specific labeling and pull down of newly synthesized RNA ($n = 4$ independent experiments). *GAPDH* levels have been used for normalization. Calculated half-lives are indicated. Bars show the mean ± SEM. Two-tailed paired Student's *t*-test was performed with *$p \leq 0.05$.

The online version of this article includes the following source data and figure supplement(s) for figure 2:

*Figure 2 continued on next page*

*Figure 2 continued*

**Source data 1.** Uncropped blots for *Figure 2A,H*.

**Source data 2.** Unedited blots for *Figure 2A, H*.

**Figure supplement 1.** Astrin is unstable in *Cluh* KO MEFs.

**Figure supplement 1—source data 1.** Uncropped blots for *Figure 2—figure supplement 1A*.

**Figure supplement 1—source data 2.** Unedited blots for *Figure 2—figure supplement 1A*.

but also at non centrosomal structures. Since lack of specific antibodies against endogenous astrin-1 precluded us to further explore this localization, we asked if endogenous CLUH can be found at the centrosome. CLUH and CEP43 stainings did not overlap in unsynchronized cells (*Figure 3E*), indicating that CLUH is not a constitutive component of the centrosome. Taken together, our results show an increased formation of the CLUH–astrin-1 complex with the progression of the cell cycle, with highest levels in S and G2 phases, and a dissolution of the complex in mitosis.

## Loss of astrin or kinastrin do not recapitulate CLUH-dependent mitochondrial phenotypes

What is the function of CLUH in complex with astrin-1 and kinastrin? We first asked whether loss of astrin or kinastrin phenocopies CLUH-dependent mitochondrial abnormalities. To avoid possible compensatory effects in stable KO clones, we employed inducible CRISPR-Cas9 astrin and kinastrin KO HeLa cells (hereafter referred to as *SPAG5* and *KNSTRN* ind-KO) in which there is constant expression of sgRNAs complementary to the target genes and the expression of the Cas9 is doxycycline inducible (*Kern et al., 2017*; *McKinley and Cheeseman, 2017*). Using this system, we induced an acute deletion of astrin and kinastrin for 4 days, in many but not all cells (*Figure 4—figure supplement 1A, B*). We focused on previously reported phenotypes observed in *CLUH* KO cells or tissues, such as mitochondrial clustering, mitochondrial fragmentation, loss of mtDNA, and decreased assembled respiratory complexes (*Gao et al., 2014*; *Schatton et al., 2017*; *Wakim et al., 2017*). Mitochondrial distribution, morphology, ultrastructure, and mtDNA levels were indistinguishable in WT, *SPAG5*, and *KNSTRN* ind-KO cells (*Figure 4A–C*, *Figure 4—figure supplement 1C–E*). Furthermore, respiratory supercomplexes containing complexes I and III assembled normally in these cells, and in-gel activity of complex I was similar to control cells (*Figure 4D*). In contrast, cells lacking CLUH displayed a reduction of assembled supercomplexes I–III, decreased activity of complex I, and abnormal mitochondrial ultrastructure (*Figure 4D*, *Figure 4—figure supplement 1E*). These data indicate that CLUH controls mitochondrial function independent from the interaction with astrin-1 or kinastrin.

## Loss of astrin rewires metabolic pathways to promote anabolism

To define how astrin and kinastrin affect cell function, we performed label-free quantitative proteomics in *SPAG5* and *KNSTRN* ind-KO cells. In line with the short time of induced KOs, only minor differences in the overall proteome in *SPAG5* and *KNSTRN* ind-KO cells compared to ind-WT cells were observed (*Figure 5A*, *Supplementary file 3*). Notably, CLUH levels were unaffected by the loss of its interactors (*Figure 5A*, *Supplementary file 3*). Astrin depletion caused a slight decrease of proteins encoded by CLUH mRNA targets (*Figure 5A*; marked in blue). Downregulated KEGG pathways, analyzed using the EnrichR webtool (using a generous cutoff of p ≤ 0.05; q ≤ 0.15) (*Chen et al., 2013*; *Kuleshov et al., 2016*; *Xie et al., 2021*), largely overlapped with metabolic pathways affected by CLUH deletion (TCA cycle, fatty acid, and amino acid degradation; *Figure 5B*; *Schatton et al., 2017*). Interestingly, metabolic pathways connected to nucleotide biosynthesis (purine and pyrimidine metabolism), the pentose phosphate pathway (PPP) and glycolysis were upregulated (*Figure 5C*). Furthermore, mTORC1 appeared as one of the most enriched terms in pathway analysis of perturbed proteins (*Figure 5—figure supplement 1A, B*).

The proteomic profile of *SPAG5* ind-KO cells is consistent with a previous study that identified astrin as a negative regulator of mTORC1 (*Thedieck et al., 2013*). We confirmed that mTORC1 is hyperactive in *SPAG5* ind-KOs upon starvation for 8 hr in HBSS (*Figure 5—figure supplement 1C, D*). To reveal if this increased signaling is reflected in cell metabolism, we performed targeted metabolomics analysis in *SPAG5* and *KNSTRN* ind-KO cells. In agreement with the mild proteomics changes, metabolomics of *SPAG5* ind-KO under basal condition did not reveal major alterations of the pools

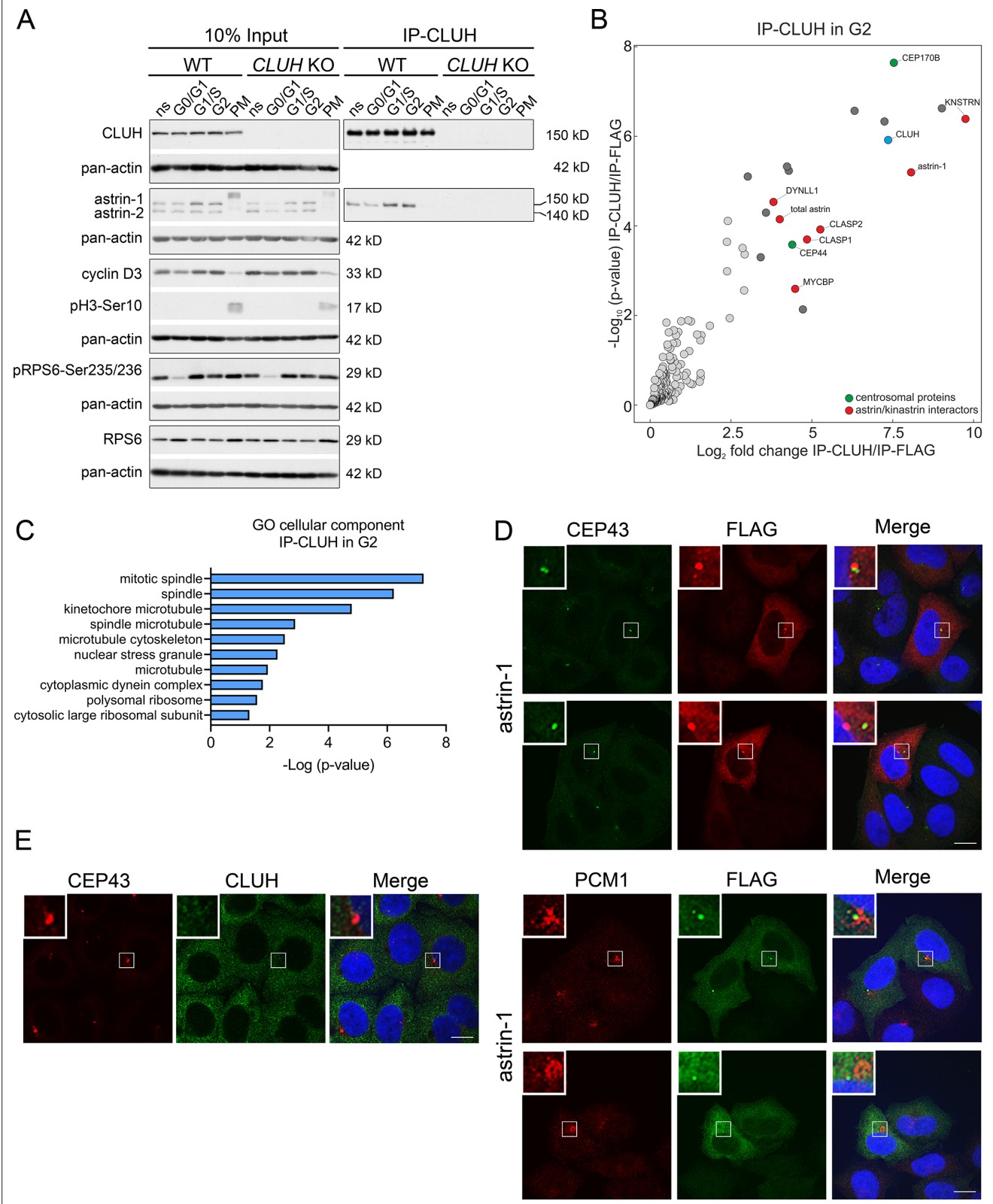

**Figure 3.** The astrin-1/CLUH complex is enriched in interphase. (**A**) Western blots of co-IPs of endogenous CLUH in WT and *CLUH* KO HeLa cells. Cells have been synchronized as shown in *Figure 3—figure supplement 1A*. Pan-actin was used as loading control and cyclin D3 as G2 phase marker, pH3-Ser10 as M phase marker and pRPS6-Ser235/236 to assess effective starvation of input samples. (**B**) Enriched proteins immunoprecipitated with an antibody against endogenous CLUH in G2-synchronized HeLa cells and detected by mass spectrometry (*n* = 4 independent replicates). Highlighted

*Figure 3 continued on next page*

*Figure 3 continued*

are all proteins enriched with a log2FC ≥ 3 and *q* ≤ 0.05. Marked in green are centrosomal proteins; marked in red are previously identified astrin/kinastrin interactors. (**C**) GO cellular component analysis of enriched proteins highlighted in B analyzed using the EnrichR webtool. (**D**) Confocal immunofluorescence pictures of HeLa cells overexpressing FLAG-tagged astrin-1 (ATG3 + 4 + 5$^{GGG}$-SPAG5) stained for CEP43 (green) and FLAG (red) or PCM1 (red) and FLAG (green). DAPI was used to stain nuclei (blue). Scale bar, 10 μm. Small boxes in left corners show a ×4 magnified area of boxed regions. (**E**) Confocal immunofluorescence pictures of HeLa cells stained for CEP43 (green) and CLUH (red). DAPI was used to stain nuclei (blue). Scale bar, 10 μm. Small boxes in left corners show a ×4 magnified area of boxed regions.

The online version of this article includes the following source data and figure supplement(s) for figure 3:

**Source data 1.** Uncropped blots for *Figure 3A*.

**Source data 2.** Unedited blots for *Figure 3A*.

**Figure supplement 1.** Controls of effective synchronization of immunoprecipitation (IP) samples.

of glycolytic and TCA cycle intermediates, amino acids, and nucleotides, with the exception of an increase in glutamine, proline, lactate, and citric acid (*Figure 5—figure supplement 2A–E*). Upon starvation, *SPAG5* depletion led to more prominent alterations of the metabolome, with accumulation of several glycolytic intermediates, lactate, and sedoheptulose-7-phosphate, in agreement with high glycolytic rate and an increase in the PPP (*Figure 5D, E*). In addition, the TCA cycle intermediates citrate, isocitrate, and malate were increased (*Figure 5F*). Both the TCA cycle and the PPP intermediates are crucial for synthesis of nucleotides, several of which accumulated in astrin-deficient cells (*Figure 5G*). Amino acid levels were unaffected (*Figure 5—figure supplement 2F*), with the exception of an increase of ornithine, a precursor of polyamines, which were also more abundant (*Figure 5H*). *KNSTRN* ind-KO cells did not show detectable metabolic differences to control cells (*Figure 5D–H*, *Figure 5—figure supplement 2A–F*). Together, our data show that cells depleted of astrin, despite starvation, engage in anabolic pathways linked to cell growth, in agreement with hyperactivated mTORC1 signaling.

## Loss of CLUH impairs anaplerotic and anabolic pathways

Increased mTORC1 signaling upon starvation is also a prerogative of CLUH-deficient cells and tissues (*Pla-Martín et al., 2020*; *Figure 5—figure supplement 1C, D*). Surprisingly, the metabolic profile of starved *CLUH* KO cells was strikingly different from that of *SPAG5* ind-KO cells (*Figure 6A–F*). Targeted metabolomics revealed a prominent decrease of the PPP intermediate sedoheptulose-7-phosphate, and a significant increase of pyruvate (*Figure 6A–C*). The intermediates of the first part of the TCA cycle (citrate, aconitate, and isocitrate) were decreased (*Figure 6C*). In addition, most amino acids that feed into the TCA cycle were decreased (*Figure 6D*), including aspartate, which is mainly produced by oxaloacetate and is an important precursor of pyrimidines (*Figure 7D*). Glutamine was instead elevated, as well as α-ketoglutarate (*Figure 7C, D*). In addition, cells lacking CLUH showed a perturbation of polyamines metabolism, with a prominent reduction of spermidine, and decreased levels of some nucleotides, especially inosine monophosphate (IMP), the first nucleotide in the synthesis pathway of purines (*Figure 6E, F*). Thus, upon starvation *CLUH* KO cells fail to maintain the TCA cycle intermediates (anaplerosis), and surprisingly show defects also in other anabolic pathways that are linked to cell growth and are regulated by mTORC1. This metabolic profile is strikingly opposite to that of *SPAG5* ind-KO cells, and indicate that the mTORC1 hyperactivation in absence of CLUH is not reflected in the cellular metabolic rewiring. This data points to an important role of CLUH in the mTORC1-dependent rewiring of cellular metabolism, by supporting mitochondrial anaplerotic pathways.

## CLUH controls cell cycle progression

In proliferating cells mitochondrial metabolism is constantly adapted to the specific metabolic needs of the different cell cycle phases. Metabolomic analyses during starvation (a treatment that arrests cell proliferation and blocks entry in G2/M) have revealed opposite signatures in cells lacking CLUH or astrin, indicating a perturbed metabolic rewiring. We therefore hypothesized that the functional significance of the interaction of CLUH with astrin is to couple mitochondrial metabolism to cell cycle progression. Assessment of unsynchronized cells using markers for different phases of the cell cycle showed a reduction of the levels of phosphorylated retinoblastoma (pRB1-Ser807/811) in absence of

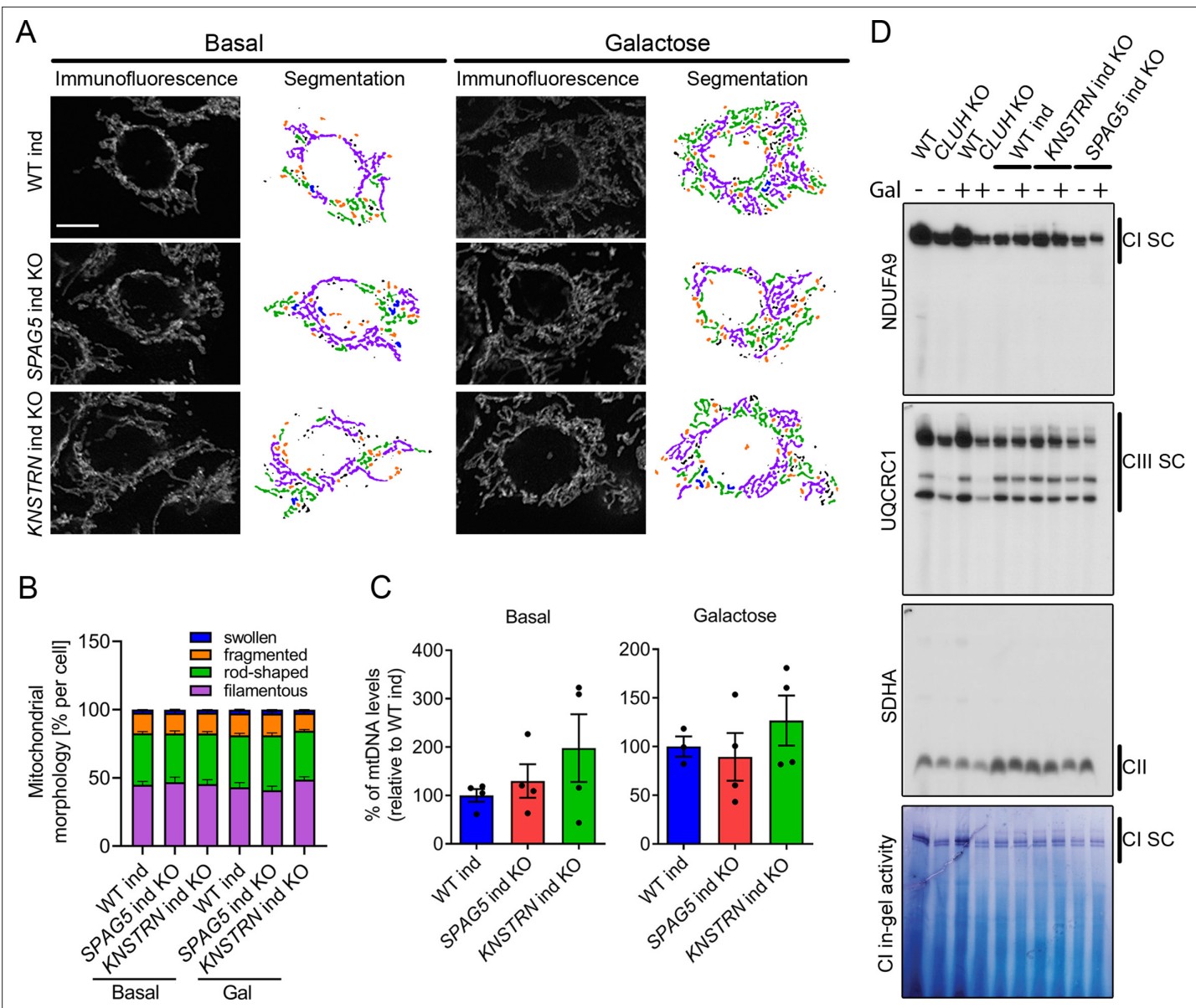

**Figure 4.** Astrin and kinastrin depletion does not mimic mitochondrial phenotypes seen upon absence of CLUH. (**A**) Mitochondrial network of WT, *SPAG5*, and *KNSTRN* ind-KO HeLa cells grown in basal or galactose media for 16 hr. Mitochondria were stained with an antibody against TOMM20. Scale bar, 10 µm. On the left side, confocal immunofluorescence pictures are shown, on the right side, mitochondrial network is shown after segmentation. Color code for different mitochondrial morphologies: purple: filamentous; green: rod-shaped; orange: fragmented; blue: swollen; black: unclassifiable. (**B**) Quantification of mitochondrial morphology of experiments as shown in A (*n* = 3 independent experiments; at least 66 in basal or 44 cells in galactose have been analyzed per genotype per replicate). Bars show the mean of each morphological class with standard error of the mean (SEM). (**C**) mtDNA levels of WT, *SPAG5*, and *KNSTRN* ind-KO HeLa cells grown in basal or galactose media for 16 hr (*n* = 4 independent experiments). Bars show mean ± SEM and dots represent values of individual replicates. (**D**) Blue native polyacrylamide gel electrophoresis (BN-PAGE) analysis of respiratory chain supercomplexes and complex I activity staining of isolated mitochondria of WT, *CLUH* KO and WT, *SPAG5*, and *KNSTRN* ind-KO HeLa cells grown in basal or galactose media for 16 hr.

The online version of this article includes the following source data and figure supplement(s) for figure 4:

**Source data 1.** Uncropped blots for *Figure 4D*.

**Source data 2.** Unedited blots for *Figure 4D*.

**Figure supplement 1.** *SPAG5* and *KNSTRN* ind KO HeLa cells show normal mitochondrial morphology.

**Figure supplement 1—source data 1.** Uncropped blots for *Figure 4—figure supplement 1A, B*.

**Figure supplement 1—source data 2.** Unedited blots for *Figure 4—figure supplement 1A, B*.

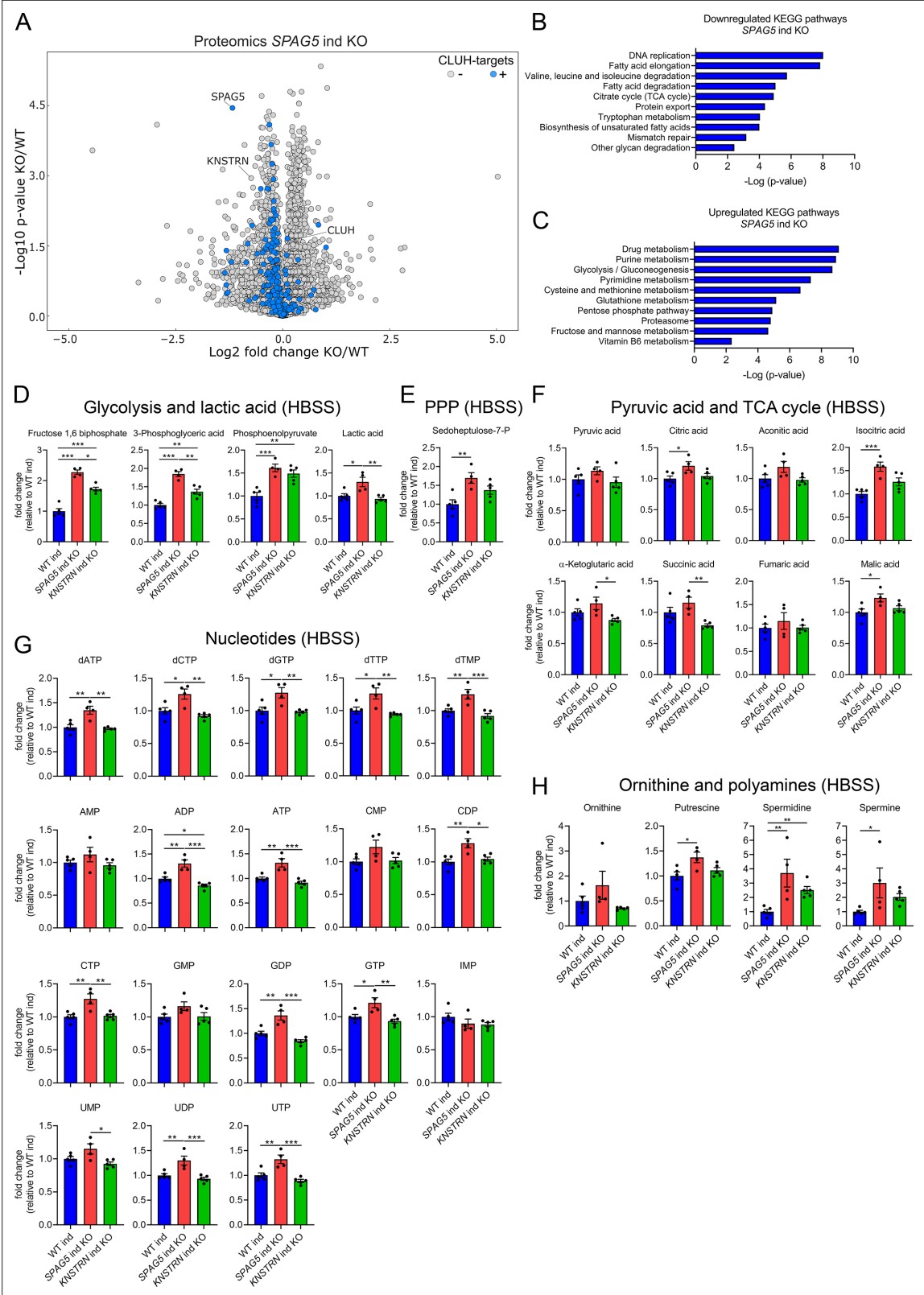

**Figure 5.** Astrin loss promotes anabolic pathways. (**A**) Volcano plot of label-free proteomics of WT and *SPAG5* ind KO HeLa cells (*n* = 4 independent replicates). CLUH targets are marked in blue. KEGG pathways of downregulated (**B**) or upregulated (**C**) proteins (with a cutoff of p ≤ 0.05; *q* ≤ 0.15) detected in proteomics analysis of *SPAG5* ind-KO cells (A and *Supplementary file 3*) using the EnrichR webtool. Targeted metabolomics of WT, *SPAG5*, and *KNSTRN* ind-KO cells after 8 hr Hanks' Balanced Salt Solution (HBSS) starvation showing glycolytic intermediates and lactic acid (**D**), sedoheptulose-

*Figure 5 continued on next page*

*Figure 5 continued*

7-P (**E**), pyruvic acid and TCA cycle intermediates (**F**), nucleotide levels (**G**), and ornithine and polyamine levels (**H**). Bars show mean ± standard error of the mean (SEM) and dots represent values of individual replicates. One-way analysis of variance (ANOVA) with post hoc Tukey's multiple comparison tests on log converted fold changes were performed with *p ≤ 0.05; **p ≤ 0.01; ***p ≤ 0.001.

The online version of this article includes the following source data and figure supplement(s) for figure 5:

**Figure supplement 1.** Loss of SPAG5 and CLUH lead to hyperactivation of mTORC1 signaling.

**Figure supplement 1—source data 1.** Uncropped blots for *Figure 5—figure supplement 1C*.

**Figure supplement 1—source data 2.** Unedited blots for *Figure 5—figure supplement 1C*.

**Figure supplement 2.** Astrin and kinastrin depletion does not affect cellular metabolism under basal conditions.

CLUH (*Figure 7A, B*). pRB1 is hyperphosphorylated during the progression through the cell cycle by cyclin D/CDK4/6. This phosphorylation releases the RB inhibitory binding to E2F transcription factors, thus allowing gene transcription required for the progression through S phase.

To further investigate the cell cycle dysregulation in *CLUH* KO cells, we used a double thymidine block (DTB) to synchronize wildtype and *CLUH* KO HeLa cells at the beginning of S phase, followed by release for different time points (*Figure 7C*). To determine the different phases of cell cycle we stained the cells with the DNA dye PI and analyzed them by flow cytometry. *CLUH* KO cells showed different dynamics in cell cycle progression compared to wildtype cells (significant changes in genotype × time in G0/G1, S, and G2/M phases; *Figure 7D–G*). During all time points, slightly more *CLUH* KO cells were in G0/G1 and less in S and G2/M phases (*Figure 7D–G*). Surprisingly, *CLUH* KO cells cycled faster than control cells, with more cells already in G2/M 3 hr after the release and more cells entering the next cell cycle at 8 hr after release (*Figure 7D–G*). In absence of CLUH, pRB1-Ser807/811 increased less during the progression of the cell cycle and peaked earlier compared to wildtype cells (*Figure 7H, I*). Cyclin D3 also showed a flattener profile in the KO compared to wildtype cells during the first 8 hr after release (*Figure 7H, I*). In contrast, the inactivating phosphorylation of CDK1 on Tyr15, which occurs in G2 and allows progression into M, and the phosphorylation of serine 10 of histone 3, which marks cells in mitosis showed a similar temporal profile to wildtype cells (*Figure 7H, I*). These results hint at the ability of *CLUH* KO cells to overcome cell cycle checkpoints at the G1/S transition. Remarkably, while CLUH levels did not change during the cell cycle, astrin-1 and to a lesser extent astrin-2 levels dropped prematurely in *CLUH* KO cells 6 hr after synchronization, a time point where most cells are in G2 (*Figure 7H, I*). These data are consistent with the increase of astrin-1 and its interaction with CLUH in G2 that we observed (*Figure 3A*).

During G1 the cells double their mass before progressing to M phase and dividing into two daughter cells. *CLUH* KO cells showed a reduced cell size at all time points during the release, in agreement with the defective growth pathways detected by the metabolomics (*Figure 7J*). Thus, upon loss of CLUH, HeLa cells escape the growth and energy checkpoint at the end of G1 and proceed faster through the cell cycle. Our results thus identify CLUH as a novel coordinator of mitochondrial activity with the cell cycle progression (*Figure 7—figure supplement 1*).

## Discussion

We have uncovered a role of CLUH in integrating mitochondrial metabolism with cell cycle progression. Mechanistically, CLUH controls the synthesis and the stability of astrin-1, the full-length protein product of the *SPAG5* gene. Astrin is a coiled-coil protein, with a well-established role during mitosis to stabilize kinetochore–microtubule interactions and allow correct orientation of chromosomes at the metaphase plate (*Manning et al., 2010*; *Dunsch et al., 2011*; *Kern et al., 2017*; *Ying et al., 2020*). Moreover, additional roles of astrin have been reported in interphase, including mTORC1 inhibition (*Thedieck et al., 2013*), centriole duplication (*Kodani et al., 2015*), and the recovery from DNA damage at the G2/M transition (*Halim et al., 2013*). We show here that the interaction of CLUH with astrin is regulated during the cell cycle, and ensures the matching of the mitochondrial metabolic output with the progression of the cell cycle.

*SPAG5* upregulation in several cancers has been associated with poor prognosis (*Yuan et al., 2014*; *Abdel-Fatah et al., 2016*; *Bertucci et al., 2016*; *Zhou et al., 2018*; *Li et al., 2019*). Human cancer cell lines show the expression of two isoforms of astrin, however the origin of these isoforms has not been

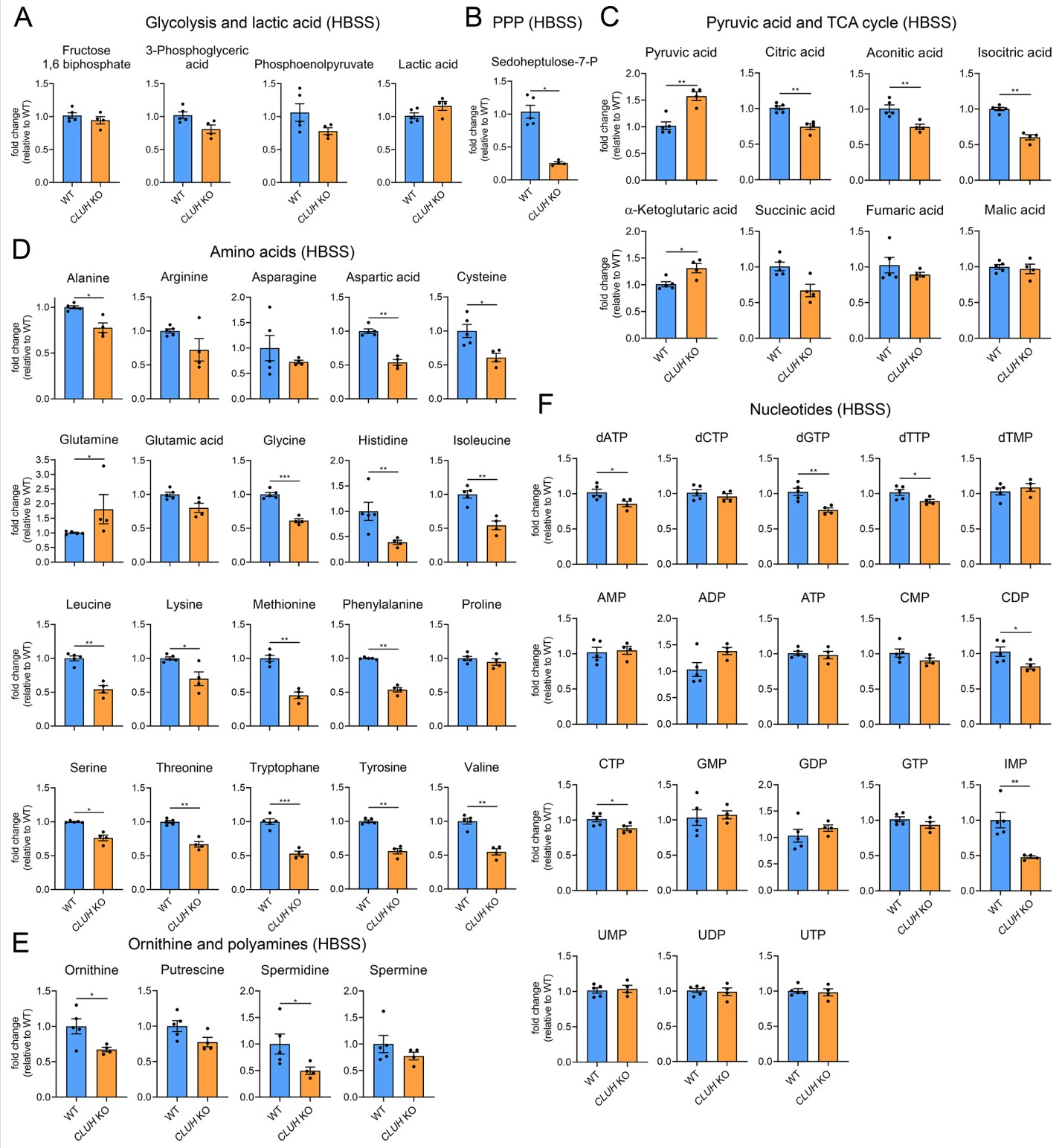

**Figure 6.** Depletion of CLUH impairs anabolic pathways. Targeted metabolomics of WT and *CLUH* KO HeLa cells after 8 hr HBSS starvation showing glycolytic intermediates and lactic acid (**A**), sedoheptulose-7-P (**B**), pyruvic acid and TCA cycle intermediates (**C**), amino acid levels (**D**), ornithine and polyamine levels (**E**), and nucleotide levels (**F**). Bars show mean ± standard error of the mean (SEM) and dots represent values of individual replicates. Two-tailed unpaired Student's *t*-tests on log converted fold changes were performed with *p ≤ 0.05; **p ≤ 0.01; ***p ≤ 0.001.

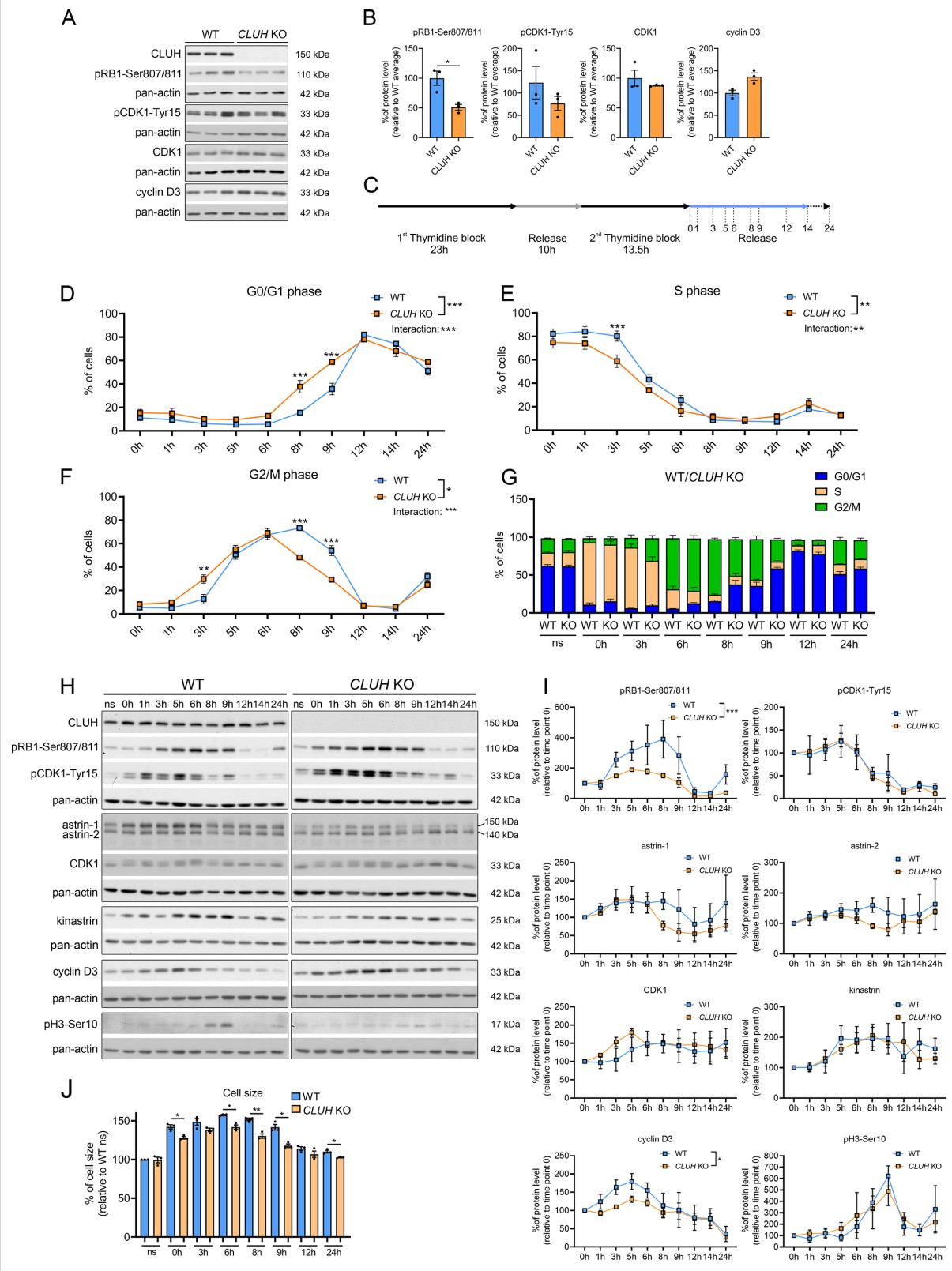

**Figure 7.** CLUH controls cell cycle progression at the G1/S boundary. (**A**) Western blots of unsynchronized WT and *CLUH* KO HeLa cells. Pan-actin was used as loading control. (**B**) Quantification of western blots shown in A (*n* = 3 independent replicates). Two-tailed paired Student's *t*-tests were performed with *p ≤ 0.05. Error bars represent standard error of the mean (SEM). (**C**) Double thymidine block (DTB) synchronization protocol used for cell cycle progression analysis. Cells were collected after release of second thymidine block at indicated time points. Percentage of WT and *CLUH* KO

*Figure 7 continued on next page*

*Figure 7 continued*

HeLa cells in G0/G1 (**D**) in S (**E**) and G2/M phase (**F**) and cell cycle distribution analysis (**G**) after DTB synchronization, collection at indicated time points and propidium iodide (PI) staining followed by flow cytometric analysis ($n$ = 3 independent experiments). For D–F, two-way analysis of variance (ANOVA) with post hoc Tukey's multiple comparison tests were performed with *$p \leq 0.05$; **$p \leq 0.01$; ***$p \leq 0.001$. Genotype × time interaction significance is also shown. Graphs and bars show mean ± SEM. (**H**) Western blots of WT and *CLUH* KO HeLa cells collected after DTB synchronization at indicated time points. Pan-actin was used as loading control. (**I**) Quantification of western blots as shown in H ($n$ = 3 independent experiments). Two-way ANOVA tests were performed with *$p \leq 0.05$; ***$p \leq 0.001$. Error bars represent SEM. (**J**) Cell size analysis of WT and *CLUH* KO HeLa cells of experiments shown in D–G. Bars show mean ± SEM and dots represent values of individual replicates. Two-tailed paired Student's *t*-tests were performed with *$p \leq 0.05$; **$p \leq 0.01$.

The online version of this article includes the following source data and figure supplement(s) for figure 7:

Source data 1. Uncropped blots for *Figure 7A,H*.

Source data 2. Unedited blots for *Figure 7A,H*.

Figure supplement 1. Model of CLUH/astrin-1 complex function in proliferating cells.

investigated up to now. We demonstrate that the two astrin isoforms arise by alternative translation initiation, and are regulated by the presence of an uORF and downstream AUGs with good Kozak sequences. Intriguingly, CLUH interacts only with the full-length astrin-1 protein. The N-terminal region of astrin is an unstructured domain that is dispensable for interaction with kinastrin, DYNLL1, MYCBP, and the kinetochore (*Kern et al., 2017*). Recently, the PLK1 kinase has been shown to interact with and phosphorylate the astrin N-terminus, an important step to stabilize the kinetochore–microtubule attachment (*Geraghty et al., 2021*). Based on our findings, we speculate that N-terminal phosphorylations are responsible for the lack of interaction of CLUH and astrin-1 in PM. Thus, astrin detaches from CLUH during mitosis when it is required for metaphase to anaphase progression.

Interestingly, CLUH appears to regulate astrin-1 expression at multiple levels. The *SPAG5* transcript represents an exception to the finding that mRNAs bound by CLUH mainly encode mitochondrial proteins (*Gao et al., 2014*). We show here that the *SPAG5* mRNA decays faster and that the synthesis of astrin-1 is impaired in absence of CLUH. It is known that during cytokinesis the E3-ubiquitin ligase MID2 ubiquitinates astrin, targeting it for proteosomal degradation (*Gholkar et al., 2016*). CLUH may be implicated in supporting synthesis of astrin-1 to replenish its levels in G1. Astrin-1 then accumulates during the S and G2 phases of the cell cycle and is stabilized, together with kinastrin, by binding CLUH. While we established the role of CLUH binding to prevent astrin-1 degradation, more experiments are required to determine to what extent the synthesis of astrin-2 is also modulated by CLUH.

The interaction of CLUH with astrin-1 led us first to investigate the possibility of a role of astrin-1 in concert with CLUH in regulating mitochondrial metabolism. Our data do not support this scenario, since depletion of astrin or kinastrin do not recapitulate the prominent mitochondrial defects observed upon CLUH downregulation or KO. Upon loss of astrin or kinastrin, mitochondria are normally dispersed in the cytoplasm, show normal morphology and ultrastructure, do not lose mtDNA and efficiently assemble respiratory supercomplexes. *SPAG5* ind-KO cells showed a mild decrease of the levels of mitochondrial proteins encoded by CLUH target mRNAs. Furthermore, starved cells lacking astrin or CLUH display strikingly opposite metabolic profiles. Finally, astrin and kinastrin are not detected in polysomal fractions, in contrast to CLUH. In agreement with our data, a recent study has also identified astrin as a CLUH partner and showed that *SPAG5* knockdown does not affect the interaction of CLUH with nuclear-encoded mitochondrial proteins occurring before their import in the organelle and revealed by Bio-ID proximity labeling (*Hémono et al., 2022*). It is possible that the formation of the astrin-1–CLUH complex competes with the function of CLUH at the polysomes. This remains a hypothesis at present and more work will be required to assess this possibility.

Our data establish an essential role of CLUH in the metabolic rewiring directed by mTORC1 signaling. Upon starvation, both *CLUH* KO and *SPAG5* ind-KO cells showed inappropriate mTORC1 activation, in agreement with astrin being a negative regulator of mTORC1 (*Thedieck et al., 2013*). It remains to be established if mTORC1 hyperactivation observed upon loss of CLUH is entirely dependent on astrin destabilization, or whether different pathways are at play. However, the metabolic defects caused by loss of CLUH were discordant with this signaling, with a decrease in the TCA cycle intermediates and in the pool of glycolytic intermediates, some nucleotides, and polyamines. Enzymes of the TCA cycle as well as characteristic anaplerotic enzymes, such as pyruvate carboxylase, proprionyl-CoA carboxylase, and other branched chain amino acid catabolic enzymes are encoded by

*bona fide* CLUH target mRNAs (*Gao et al., 2014*). These pathways are essential to replenish the TCA cycle intermediates to compensate for their loss upon activation of biosynthetic pathways following mTORC1 signaling. Indeed, a compensatory increase in anaplerotic pathways occurs under conditions of chronic mTORC1 activation (*Dutchak et al., 2018*). Therefore, CLUH tunes mTORC1 signaling in two independent ways, by regulating the expression of astrin-1 and by ensuring the maintenance of mitochondrial anaplerotic pathways via its RNA-binding function (*Figure 7—figure supplement 1*). Our data demonstrate another connection between mTORC1 signaling and mitochondrial function. mTORC1 positively regulates the translation of mitochondrial proteins involved in oxidative phosphorylation and mitochondrial dynamics (*Morita et al., 2013*; *Morita et al., 2017*), while inhibition of mTORC1 activates a lipid signaling cascade that triggers YME1L-mediated intramitochondrial proteolysis to limit mitochondrial biogenesis (*MacVicar et al., 2019*). Furthermore, mTORC1 hyperactivation is part of the integrated stress response that follows mitochondrial dysfunction (*Khan et al., 2017*).

Proliferating cells must shunt metabolites into mitochondrial pathways that promote cell growth or oxidative respiration and ATP production, depending on the cell cycle phase (*DeBerardinis et al., 2008*; *Salazar-Roa and Malumbres, 2017*). In G1 cells double their mass and depend on glycolysis to fuel the TCA cycle and sustain biosynthetic pathways, while in S synthesis of nucleotides is required and glutamine oxidation is prevalent. Interestingly, the TCA cycle intermediates fluctuate during the cell cycle, and it is known that these variations are not transcriptionally regulated (*Olsen et al., 2010*; *Ahn et al., 2017*). In mammalian cells, mTORC1 activation is important for the transition from G1 to S, an important check point when the energy levels and the growth status of the cell are sensed, and from G2 to M (*Cuyàs et al., 2014*). We hypothesize that by regulating astrin-1 and stabilizing an RNA regulon involved in the TCA cycle and anaplerotic pathways (*Gao et al., 2014*), CLUH ensures that the progression of the cell cycle is adjusted with activation of mTORC1 signaling and the mitochondrial metabolic profile (*Figure 7—figure supplement 1*). Using a post-transcriptional mechanism to coordinate mitochondrial function with cell cycle phases allows a fast and flexible mean to control a broad gene program. Consistently, cells lacking CLUH have cell cycle defects, characterized by a faster cell cycle progression despite failure to properly double the cell mass.

Astrin is a component of centriolar satellites in interphase, and it is required for the centrosomal localization of CDK5RAP2, a protein involved in microcephaly, and for proper centriole duplication in S phase (*Thein et al., 2007*; *Kodani et al., 2015*), a step which is crucial for cell cycle progression. Astrin-1 overexpression depletes endogenous CLUH from the cytoplasm and recruits it to structures that are close to centrosomes or contain centrosomal markers. However, we could not demonstrate the recruitment of endogenous CLUH at the centrosome at least in unsynchronized cells. Whether and how astrin-1 is involved in regulating CLUH subcellular localization remains to be determined.

Finally, our data add to the evidence that astrin plays additional roles outside mitosis. Astrin was also identified in an unbiased screen to be essential for the reentry in mitosis after DNA damage (*Halim et al., 2018*). This interphase role of astrin should be further explored in view of our findings, as possibly involving dysregulated or enhanced CLUH activity. HeLa cells, as many human cancer cells, overexpress not only astrin-1 but also astrin-2, an isoform of astrin competent to bind all other interactors (kinastrin, DYLNN1, and MYCPB) and the kinetochore (*Kern et al., 2017*), but not CLUH. We propose that expression of astrin-2 endows cancer cells with the ability to proliferate, despite a dysregulated coupling between mitochondrial metabolism and the cell cycle, making CLUH a possible novel therapeutic target in cancers overexpressing *SPAG5*. In conclusions, our findings reveal a novel post-transcriptional mechanism coordinating mitochondrial metabolism and the cell cycle.

## Materials and methods
### Cell lines
HeLa WT and *CLUH* KO cells were obtained from Guy Lenaers lab (Université d'Angers) and previously described (*Wakim et al., 2017*). Doxycycline inducible CRISPR-Cas9 HeLa WT, *SPAG5*, and *KNSTRN* KO cells (*Kern et al., 2017*; *McKinley and Cheeseman, 2017*) were kindly provided by Iain Cheeseman (Whitehead Institute for Biomedical Research, Cambridge, USA). HeLa cells have not been further validated, but in all cases parental HeLa cells have been used as controls. Immortalized WT and *Cluh* KO MEFs were previously described (*Gao et al., 2014*). HEK293T cells were purchased from a vendor several years ago and not further validated. These cells were used to stably express

inducible 3xFLAG-kinastrin and employed only for a biochemical experiment which was confirmed with independent cell lines. To this end, full-length N-terminally 3xFLAG-tagged human *KNSTRN* ORF was cloned into pTREx-DEST30 vector by serial Gateway recombinations (Invitrogen) according to the manual. Stable transfected HEK293T cells were generated with the Flp-In T-REx 293 cell system (Invitrogen) following the manual. Positive cells were selected with media containing 1.5 mg/ml hygromycin (InvivoGen) and 150 µg/ml blasticidin (InvivoGen) starting 24 hr after transfection and single colonies were picked to generate a monoclonal cell line. All cell lines were regularly checked for *Mycoplasma* contamination and found negative.

## Cell culture

All cells were cultured at 37°C and 5% constant $CO_2$ supply. All cell culture media and ingredients were purchased from Gibco unless stated otherwise. HeLa cells and MEFs were grown in Dulbecco's modified eagle medium (DMEM) including 4.5 g/l glucose supplemented with 2 mM L-glutamine, 2% penicillin (10,000 u/ml)/streptomycin (10,000 µg/ml) and 10% FetalClone III serum (Hyclone, Thermo Fisher Scientific). HEK293T cells were cultured in DMEM including 4.5 g/l glucose supplemented with 2 mM L-glutamine, 1 mM sodium pyruvate, 1% nonessential amino acids, 10% tetracycline-free fetal bovine serum (Biochrom AG), 1.5 mg/ml hygromycin, and 150 µg/ml blasticidin. To induce overexpression, cells were treated with 1 µg/ml tetracycline (Sigma-Aldrich) for 16 hr. Doxycycline inducible CRISPR-Cas9 HeLa WT, *SPAG5* and *KNSTRN* KO cells were grown in DMEM containing 4.5 g/l glucose supplemented with 2 mM L-glutamine, 2% penicillin (10,000 u/ml)/streptomycin (10,000 µg/ml), and 10% tetracycline free fetal bovine serum (Sigma-Aldrich). To induce Cas9 expression, cells were treated with 1 µg/ml doxycycline (Sigma-Aldrich) for 4 consecutive days adding fresh doxycycline each day. To obtain stable *SPAG5* and *KNSTRN* KO cell lines, induced cells underwent monoclonal selection after serial dilutions.

## Metabolic labeling

For metabolic labelling of newly synthesized astrin, cells were primed in metabolic labeling medium [DMEM containing 4.5 g/l glucose without L-methionine and L-cystine (#21013024, Gibco) including 10% dialysed serum, 2% penicillin (10,000 u/ml)/streptomycin (10,000 µg/ml), 2 mM L-glutamine, 1 mM sodium pyruvate, and 1% nonessential amino acids] for 30 min followed by incubation with metabolic labeling medium including $^{35}$S-methionine (50 µCi per 10 cm plate) for indicated time points.

## SILAC labeling

Cells were cultured in DMEM medium without glutamine, arginine, and lysine (Silantes, # 280001200), supplemented with 2 mM L-glutamine, 2% penicillin/streptomycin, dialyzed FCS (Thermo Fisher, # 26400044), 28 µg/ml L-arginine-HCl (Arg0 or Arg10) (Silantes, # 201604102), and 73 µg/ml L-lysine-2HCl (Lys0 or Lys8) (Silantes, # 211604102) for three passages before collecting for IP.

## CHX chase and galactose treatment

To assess protein stability, cells were treated with 0.1 mg/ml CHX (Sigma-Aldrich) with or without 20 µM MG132 (Sigma-Aldrich) for indicated time points. Cells were grown when indicated in galactose media [DMEM, no glucose (#11966-025, Gibco) supplemented with 10 mM galactose (Sigma-Aldrich), 2 mM L-glutamine, 1 mM sodium pyruvate, 2% penicillin (10,000 u/ml)/streptomycin (10,000 µg/ml) and 10% dialyzed fetal bovine serum (Gibco)] for 16 hr before experiments were performed.

## Cloning and mutagenesis of SPAG5 constructs

The underlying sequence of human *SPAG5* used in this study can be found under the accession number NM_006461.3 on the NCBI database. Human full-length *SPAG5* ORF (ATG1-SPAG5) or excluding the first 453 nucleotides (Δ151-SPAG5) were cloned into p3xFLAG-CMV-14 (Sigma-Aldrich) using NotI or NotI/ClaI restriction sites, respectively. To obtain a construct including the 5′ UTR (5UTR-SPAG5), the 5′ UTR of *SPAG5* was cloned into Δ151-SPAG5-FLAG using HindIII restriction site. To mutagenize ATGs to GGG, DpnI site directed mutagenesis was employed. Mutagenized base pair positions and construct names are indicated in *Figure 1C* and *Figure 1—figure supplement 1A*.

## Cell transfection

Cells were transfected with Lipofectamine 2000 (Invitrogen) in a plasmid to transfection reagent ratio of 1:5 according to the manual. After 24 hr of overexpression, cells were harvested by scraping and pelleting for lysis or fixed with respective reagent for immunofluorescence. RNA interference of *CLUH* was done as described before (*Gao et al., 2014*). Briefly, HeLa cells were transfected with 100 nM of siRNA against human *CLUH* or control siRNA using Lipofectamine 2000 according to the instructions of the manual and experiments were performed after 72 hr of downregulation.

## Polysome profiling

Cells were grown in 15 cm dishes to 70–80% confluency and treated for 15 min with fresh media supplied with 100 µg/ml CHX (Sigma Aldrich), followed by crosslinking with 1 mM dithiobis (succin-imidyl propionate) in phosphate-buffered saline (PBS) for 30 min at RT prior to polysome profiling followed by immunoblotting. Cells were washed with PBS and quenched in 20 mM Tris–HCl, pH 7.4, 5 mM L-cysteine for 10 min. Cells were then washed twice with ice cold PBS containing 100 µg/ml CHX and scraped in 1.5 ml ice cold PBS including CHX and collected in a 2 ml tube. Cells were immediately centrifuged at 21,000 × g for 10 s at 4°C and supernatant was discarded. Cells were lysed for 30 min on ice in buffer comprised of 20 mM Tris–HCl, pH 7.4, 30 mM KCl, 15 mM MgCl₂, 0.5% Triton X-100 (vol/vol), 2 mM Dithiothreitol (DTT), 1 mg/ml heparin, 100 µg/ml CHX, 0.16 U/ml RNase inhibitor (RNasin Plus; Promega), and 1× Ethylenediaminetetraacetic acid (EDTA)-free protease cocktail (Roche). Cell debris were then removed with 5 min centrifugation at 14,000 × g at 4°C and protein concentration was measured with the standard Bradford assay. Cell lysate was then applied on a continuous 7–47% sucrose gradient (mol weight/volume) in ultra clear tubes (Beckman & Coulter, #331372) and centrifuged at 97,658 × g for 3 hr at 4°C using a SW41Ti rotor (Beckman & Coulter, # 331362). The polysome fractions were collected using the Foxy R1 Fraction Collector and immediately snap frozen with liquid nitrogen and stored at −80°C. The polysome profile was detected with the UA-6 detector (Teledyne ISCO) during the collection of each polysome fraction.

## Synchronization of cells

Cells were synchronized by DTB using an adapted protocol (*Dai et al., 2018*). Briefly, cells were plated in desired amount and treated the next day with 2 mM thymidine (Sigma-Aldrich) for 23 hr, afterwards cells were released in standard media without thymidine for 10 hr and second thymidine block (2 mM thymidine) was performed for 13.5 hr. Next day, cells were released and collected at indicated time points depending on the experiment (*Figure 3—figure supplement 1A*, *Figure 7C*). To enrich cells in PM phase, cells were treated with 100 ng/µl nocodazole (Sigma-Aldrich) after 3 hr of release and collected after 12 additional hours. To block cells in G0/G1, cells were starved in media without serum for 16 h.

## Immunoprecipitations

For IP cells were collected and lysed in an appropriate volume of IP buffer [50 mM Tris–HCl, pH 7.4; 50 mM KCl; 0.1% Triton X-100 supplemented freshly with protease inhibitor cocktail (Sigma-Aldrich)] for 30 min on ice after passing 3× through syringe (30G × 1/2″, B. Braun Sterican). For SILAC samples, the lysate was incubated with 25 U benzonase HC nuclease (Sigma-Aldrich) at 37°C for 30 min before IP. Afterwards, lysates were cleared by centrifugation at 20,000 × g for 30 min and protein amount was determined by standard Bradford assay (BioRad). For each reaction, 300–500 µg of protein were diluted in 250 µl IP buffer and incubated for 3 hr in head-to-toe agitation at 4°C with 0.5 µg of the specific antibody: rabbit polyclonal rabbit anti-CLUH [#NB100-93305 (1) and #NB100-93306 (2) from Novus Biologicals]; rabbit polyclonal anti-kinastrin (#SAB1103031 from Sigma-Aldrich); rabbit poly-clonal anti-astrin (#14726-1-AP from ProteinTech; #NB100-74638 from Novus Biologicals). As control antibodies, we used rabbit polyclonal anti-AFG3L1 (*Koppen et al., 2007*; *Figure 1A, B*, *Figure 1—figure supplement 1C*) and rabbit polyclonal anti-FLAG (#F7425 from Sigma-Aldrich) (*Figure 3*, *Figure 3—figure supplement 1B, D*). 20 µl of prewashed magnetic Dynabeads Protein G (Invitrogen) were added per reaction and incubated for 1 hr in head-to-toe agitation at 4°C. Afterwards, beads were washed five times with IP buffer. To elute kinastrin, beads were incubated in 100 mM glycine, pH 2.3 for 20 min at 4°C. In the other cases, proteins were eluted in 30 µl 3× Laemmli buffer (20 mM Tris–HCl, pH 6.8, 2% SDS, 5% β-mercaptoethanol, 2.5% glycerol, and 2.5% bromophenol blue) by

vortexing for 1 min and boiling at 95°C for 5 min. Samples were stored at −20°C or run immediately on SDS–PAGE.

## Sample preparation for mass spectrometry

For IP of CLUH followed by mass spectrometry of WT cells enriched in G2, experiments were carried out as described before using 400 µg lysate as input and elution was done with 30 µl SP3 lysis buffer (5% SDS in 1× PBS) by vortexing for 1 min and boiling at 95°C for 5 min. Afterwards, proteins were reduced with 5 mM dithiothreitol for 30 min at 55°C and alkylated with 40 mM chloroacetamide at RT for 30 min in the dark. Next, samples were centrifuged at 20,000 × *g* for 10 min and supernatant was transferred to new tube and stored at −20°C before mass spectrometry was performed. For SILAC labeled samples, beads were resuspended in 50 µl of elution buffer 1 (2 M urea, 50 mM triethylammoniumbicarbonate, 1 mM DTT, 5 ng/µl trypsin) and incubated at RT for 30 min while shaking. Beads were centrifuged and supernatant transferred to a new tube. Beads were washed twice with elution buffer 2 (2 M urea, 50 mM triethylammoniumbicarbonate, 5 mM chloroacetamide) and centrifuged. The eluates were combined. Next, proteins were digested with lysyl endopeptidase (Wako Pure Chemical Industries) and trypsin (Sigma-Aldrich) with an enzyme:substrate ratio of 1:75 at 37°C for 16 hr. Next day, samples were acidified with formic acid to stop enzymatic digestion, purified with and loaded on StageTips as described before (*Rappsilber et al., 2007*).

HeLa WT, *SPAG5*, and *KNSTRN* ind-KO cells were collected by scraping, pelleted and resuspended in appropriate amount of lysis buffer [50 mM Tris–HCl, pH 7.4; 150 mM NaCl; 1 mM EDTA, pH 8; 1% IGEPAL CA-630; 0.25% sodium deoxycholate freshly supplemented with protease inhibitor cocktail (Sigma-Aldrich)]. Lysates were passed 3× through syringe (30G × 1/2″, B. Braun Sterican) and incubated on ice for 30 min. After centrifugation for 30 min at 20,000 × *g* at 4°C, protein amounts were determined using standard Bradford assay (BioRad) and 30 µg of protein lysate were precipitated with acetone. Briefly, 4× volume of ice-cold acetone were added to lysates, incubated for 15 min at −80°C followed by 90 min incubation at −20°C and centrifugation for 15 min at 16,000 × *g*. Pellets were washed in ice-cold acetone, air-dried and resuspended in 50 µl of 8 M urea in 50 mM triethylammoniumbicarbonate including protease inhibitor cocktail (Roche). Afterwards proteins were reduced with 5 mM dithiothreitol for 1 hr at 25°C and alkylated with 40 mM chloroacetamide for 30 min in the dark. Next, proteins were digested with lysyl endopeptidase (Wako Pure Chemical Industries) with an enzyme:substrate ratio of 1:75 at 25°C for 4 hr. Samples were diluted with 50 mM triethylammoniumbicarbonate to reach a final urea concentration of 2 M. Then proteins were digested with trypsin (Sigma-Aldrich) with an enzyme:substrate ratio of 1:75 and incubation at 25°C for 16 hr. Next day, samples were acidified with formic acid to stop enzymatic digestion, purified with and loaded on StageTips as described before (*Rappsilber et al., 2007*). Samples were stored in dried StageTips at 4°C until mass spectrometry was performed.

## Mass spectrometry of CLUH IP after SILAC

SILAC labeled samples were analyzed on a Q Exactive Plus Orbitrap (Thermo Scientific) mass spectrometer that was coupled to an EASY nLC (Thermo Scientific). Peptides were loaded with solvent A (0.1% formic acid in water) onto an in-house packed analytical column (50 cm × 75 µm I.D., filled with 2.7 µm Poroshell EC120 C18, Agilent). Peptides were chromatographically separated at a constant flow rate of 250 nl/min using the following gradient: 7–23% solvent B (0.1% formic acid in 80% acetonitrile) within 35.0 min, 23–32% solvent B within 5.0 min, 32–85% solvent B within 5.0 min, followed by washing and column equilibration. The mass spectrometer was operated in data-dependent acquisition mode. The MS1 survey scan was acquired from 300 to 1750 *m/z* at a resolution of 70,000. The top 10 most abundant peptides were isolated within a 1.8 Th window and subjected to HCD fragmentation at a normalized collision energy of 27%. The AGC target was set to 5e5 charges, allowing a maximum injection time of 108ms. Product ions were detected in the Orbitrap at a resolution of 35,000. Precursors were dynamically excluded for 20.0 s. All mass spectrometric raw data were processed with Maxquant (version 1.5.3.8) using default parameters. Briefly, MS2 spectra were searched against the Uniprot HUMANc_UP000005640.fasta (downloaded at: 26.08.2020) database, including a list of common contaminants. False discovery rates (FDR) on protein and peptide-to-spectrum match (PSM) level were estimated by the target-decoy approach to 1% (Protein FDR) and 1% (PSM FDR), respectively. The minimal peptide length was set to seven amino acids and carbamidomethylation at

cysteine residues was considered as a fixed modification. Oxidation (M) and acetyl (Protein N-term) were included as variable modifications. SILAC/dimethyl labeling quantification was used, and the requantify option was enabled.

## Mass spectrometry of CLUH IP in synchronized cells

Immunoprecipitated proteins from cells synchronized in G2 were analyzed on a Q-Exactive Plus (Thermo Scientific) mass spectrometer that was coupled to an EASY nLC 1200 UPLC (Thermo Scientific). Peptides were loaded with solvent A (0.1% formic acid in water) onto an in-house packed analytical column (50 cm × 75 µm I.D., filled with 2.7 µm Poroshell EC120 C18, Agilent). Peptides were chromatographically separated at a constant flow rate of 250 nl/min using the following gradient: 3–5% solvent B (0.1% formic acid in 80% acetonitrile) within 1 min, 5–30% solvent B (0.1% formic acid in 80% acetonitrile) within 40 min, 30–50% solvent B 8 min and 40–95% solvent B within 1 min, followed by washing with 95% solvent B for 10 min. The mass spectrometer was operated in data-dependent acquisition mode. The MS1 survey scan was acquired from 300 to 1750 *m/z* at a resolution of 70,000. The top 10 most abundant peptides were isolated within a 1.8 Th window and subjected to HCD fragmentation at a normalized collision energy of 27%. The AGC target was set to 5e5 charges, allowing a maximum injection time of 110 ms. Product ions were detected in the Orbitrap at a resolution of 35,000. Precursors were dynamically excluded for 10 s. All mass spectrometric raw data were processed with MaxQuant version 1.5.3.8 (*Tyanova et al., 2016b*) using default parameters. Briefly, MS2 spectra were searched against a canonical Uniprot human fasta database, which was modified by replacing the default entry for SPAG5 (Q96R06) by two separate entries representing (1) the N-terminal 125 amino acids and (2) the C-terminal sequence from position 126 on. The MaxQuant default list was used to filter for common contaminants. False discovery rates on protein and PSM level were estimated by the target-decoy approach to 1% (Protein FDR) and 1% (PSM FDR), respectively. The minimal peptide length was set to seven amino acids and carbamidomethylation at cysteine residues was considered as a fixed modification. Oxidation (M) and Acetyl (Protein N-term) were included as variable modifications. The match-between runs option was restricted to replicates of the same condition. LFQ quantification was used with default settings. LFQ intensities were loaded into in Perseus version 1.6.1.1 (*Tyanova et al., 2016a*). Decoys and potential contaminants were removed and the dataset was filtered for at least 4 out of 4 values in at least one condition. Remaining missing values were imputed with random values from the left end of the intensity distribution using Perseus defaults. Two sample Student's *t*-tests were calculated using permutation-based FDR estimation.

## Mass spectrometry of *SPAG5* and *KNSTRN* ind-KO cells

For proteomics of HeLa WT ind, *SPAG5*, and *KNSTRN* ind-KO cells, peptide digests were analyzed on a Q Exactive plus Orbitrap (Thermo Scientific) mass spectrometer that was coupled to an EASY nLC (Thermo Scientific). Samples were loaded onto an in-house packed analytical column (50 cm × 75 µm I.D., filled with 2.7 µm Poroshell EC120 C18, Agilent). Peptides were separated at a flow rate of 250 nl/min and the following gradient: 3–5% solvent B (0.1% formic acid in 80% acetonitrile) within 1.0 min, 5–30% solvent B within 91.0 min, 30–50% solvent B within 17.0 min, 50–95% solvent B within 1.0 min, followed by washing with 95% solvent B for 10 min. DDA runs for spectrum library generation were acquired from distinct pools of the sample groups and Hek293 cell digests fractionated high pH HPLC. MS1 survey scan were acquired at a resolution of 70,000. The top 10 most abundant peptides were isolated within a 2.0 Th window and subjected to HCD fragmentation with normalized collision energy of 27%. The AGC target was set to 5e5 charges, allowing a maximum injection time of 105 ms. Product ions were detected in the orbitrap at a resolution of 35,000. Precursors were dynamically excluded for 20.0 s. Sample runs were acquired in data-independent mode using 10 variable windows covering the mass range from *m/z* 450 to *m/z* 1200. MS1 scans were acquired at 140,000 resolution, maximum IT restricted to 120 ms and an AGC target set to 5e6 charges. The settings for MS2 scans were 17,500 resolution, maximum IT restricted to 60 ms and AGC target set to 5e6 charges. The default charge state for the MS2 was set to 4. Stepped normalized collision energy was set to 23.5, 26, and 28.5. All spectra were acquired in profile mode. A hybrid spectrum library was generated in Spectronaut 13 (*Bruderer et al., 2015*) using DDA library runs, DIA sample runs and a canonical human sequence file (SwissProt, 20,416 entries) downloaded from Uniprot. Spectronaut default settings were used for the analysis of the DIA runs. Protein identifications were filtered for *q* values below 0.01 and

normalized intensities were exported for subsequent statistical analysis in Perseus 1.6.1.1 (*Tyanova et al., 2016b*). Intensities were transformed to log2 values and the dataset was filtered for at least 4 out of 4 values in at least one condition. Remaining missing values were imputed with random values from the left end of the intensity distribution (with 0.3 sd, downshift 2 sd). Two sample Student's *t*-tests were calculated using permutation-based FDR estimation.

## Proteomics visualization and pathway analysis

Enriched proteins of IP experiment or proteomics results were visualized as volcano plots using Instant Clue software (*Nolte et al., 2018*) and pathway analysis was carried out using the EnrichR webtool with a cutoff of $q \leq 0.05$ and log2 fold change $\geq 3$ for the IP and a cutoff of $q \leq 0.15$ and $p \leq 0.05$ for the proteomics analysis (*Chen et al., 2013*; *Kuleshov et al., 2016*; *Xie et al., 2021*).

## Isolation of mitochondria

Cells were collected from confluent 15 cm plates with trypsinization, washed twice with PBS, and resuspended in an ice-cold mitochondria isolation buffer containing 20 mM HEPES, pH 7.6; 220 mM mannitol; 70 mM sucrose; 1 mM EDTA; 0.2% fatty acid-free bovine serum albumin (BSA). After 20 min of incubation on ice, cells were homogenized using the rotational engine homogenizer (Potter S, Sartorius; 30 strokes, 1200 rpm) followed by centrifugation at $850 \times g$. Next, mitochondria were pelleted at $8500 \times g$ for 10 min at 4°C, washed with BSA-free buffer, and protein concentration was determined with Bradford reagent (Sigma-Aldrich). For further analysis, mitochondria were subjected to blue native polyacrylamide gel electrophoresis (BN-PAGE) followed by western blotting or determination of the in-gel activity of respiratory complexes.

## Analysis of mitochondrial respiratory complexes with BN-PAGE

20 mg of mitochondria were lysed with digitonin (Calbiochem; 6.6 g/g protein) for 15 min on ice with occasional vortexing and cleared from insoluble material for 20 min at $20,000 \times g$ at 4°C. Lysates were combined with Coomassie G-250 (0.25% final concentration). Mitochondrial respiratory supercomplexes were resolved with BN-PAGE using the 4–16% NativePAGE Novex Bis-Tris Mini Gels (Invitrogen) in a Bis-Tris/Tricine buffering system with cathode buffer initially supplemented with 0.02%G-250 followed by the 0.002% G-250. For *complex I in-gel activity,* gels were incubated at RT in a buffer containing 0.01 mg/ml NADH and 2.5 mg/ml nitrotetrazolium blue in 5 mM Tris–HCl, pH 7.4.

## Cell lysis and western blot

Cell pellets were lysed in appropriate amount of lysis buffer [50 mM Tris–HCl, pH 7.4; 150 mM NaCl; 1 mM EDTA, pH 8; 1% IGEPAL CA-630; 0.25% sodium deoxycholate freshly supplemented with protease inhibitor cocktail (Sigma-Aldrich)] and protein amounts were determined by standard Bradford assay (BioRad). Desired protein amounts were mixed with appropriate volume of 3× loading buffer (20 mM Tris–HCl, pH 6.8; 2% SDS; 5% β-mercaptoethanol; 2.5% glycerol, and 2.5% bromophenol blue), boiled for 5 min at 95°C and loaded on SDS polyacrylamide gels. Proteins were separated by SDS–PAGE and blotted on polyvinylidene fluoride (PVDF) membranes using wet transfer. After BN-PAGE, separated mitochondrial complexes were transferred on PVDF membranes using the wet transfer sodium lauryl sulfate (SDS)- and methanol-free system. The following primary antibodies were used for western blotting: rabbit polyclonal anti-CLUH antibodies [detecting human CLUH; #NB100-93305 (1), #NB100-93306 (2)], rabbit polyclonal anti-RPS6 (#NB100-1595), rabbit polyclonal anti-RPL7 (#NB100-2269) antibodies from Novus Biologicals; rabbit polyclonal anti-astrin (#14726-1-AP) antibody from ProteinTech; rabbit polyclonal anti-FLAG (#F7425) and mouse monoclonal anti-FLAG (#F3165) antibodies from Sigma-Aldrich; mouse monoclonal pan-actin (#MAB1501) and anti-GAPDH (#MAB374) antibodies from EMD Millipore; rabbit polyclonal anti-CLUH antibody (detecting murine CLUH; #ARP70642_P050) from Aviva; rabbit polyclonal anti-kinastrin (#ab122769) and rabbit polyclonal pH3-Ser10 (#ab5176) antibodies from Abcam; mouse monoclonal anti-SDHA (#459200), anti-NDUFA9 (#459100), and anti-UQCRC1 (#459140) from Molecular probes; rabbit polyclonal pRB1-Ser807/811 (#9308), rabbit monoclonal pCDK1-Tyr15 (#4539), mouse monoclonal anti-cyclin D3 (#2936), and rabbit polyclonal pRPS6-Ser235/236 (#2211) antibodies from Cell Signaling and mouse monoclonal anti-CDK1 (#sc-54) antibody from Santa Cruz Biotechnologies.

## Sample collection to measure anionic metabolites, amino acids, and polyamines

WT, *SPAG5*, and *KNSTRN* ind-KO cells were induced as described before and 1,000,000 cells were plated in 6-well plates the day before the extraction. Next day, the cells were either collected immediately (basal condition) or starved for 8 hr in HBSS media (#14025092, Gibco) in absence of doxycycline. Cells were washed twice with buffer containing 75 mM ammonium carbonate (pH 7.4) and metabolites were extracted with cold (−20°C) extraction solvent (40:40:20 acetonitrile:methanol:water) and incubation for 10 min at −20°C. Supernatant was collected and extraction was repeated. Afterwards, cells were scraped on ice and combined with the supernatant of the previous step. Samples were immediately dried in a speed vac concentrator and dried pellets were kept at −80°C until mass spectrometry was performed.

## Anion-exchange chromatography mass spectrometry for the analysis of anionic metabolites

Extracted metabolites were resuspended in 150 µl of Optima LC/MS grade water (Thermo Fisher Scientific), of which 100 µl were transferred to polypropylene autosampler vials (Chromatography Accessories Trott, Germany) before anion-exchange chromatography mass spectrometry analysis. The samples were analysed using a Dionex ion chromatography system (Integrion, Thermo Fisher Scientific) as described previously (*Schwaiger et al., 2017*). In brief, 5 µl of polar metabolite extract were injected in push partial mode using an overfill factor of 3, onto a Dionex IonPac AS11-HC column (2 mm × 250 mm, 4 µm particle size, Thermo Fisher Scientific) equipped with a Dionex IonPac AG11-HC guard column (2 mm × 50 mm, 4 µm, Thermo Fisher Scientific). The column temperature was held at 30°C, while the auto sampler was set to 6°C. A potassium hydroxide gradient was generated using a potassium hydroxide cartridge (Eluent Generator, Thermo Scientific), which was supplied with deionized water. The metabolite separation was carried at a flow rate of 380 µl/min, applying the following gradient conditions: 0–3 min, 10 mM KOH; 3–12 min, 10–50 mM KOH; 12–19 min, 50–100 mM KOH; 19–21 min, 100 mM KOH; 21–22 min, 100–10 mM KOH. The column was re-equilibrated at 10 mM for 8 min. For the analysis of metabolic pool sizes the eluting compounds were detected in negative ion mode [M−H]$^-$ using multiple reaction monitoring mode with the following settings: capillary voltage 2.7 kV, desolvation temperature 550°C, desolvation gas flow 800 l/hr, collision cell gas flow 0.15 ml/min. The detailed quantitative and qualitative transitions and electronic settings for the analyzed metabolites are summarized in *Supplementary file 4*. The MS data analysis was performed using the TargetLynx Software (Version 4.1, Waters). For data analysis, the area of the quantitative transition of each compound was extracted and integrated using a retention time (RT) tolerance of <0.1 min as compared to the independently measured reference compounds. Areas of the cellular pool sizes were normalized to the internal standards (citric acid D4), which were added to the extraction buffer, followed by a normalization to the protein content of the analyzed sample. One sample of *CLUH* KO and one of *SPAG5* ind-KO cells upon HBSS starvation have been classified as outlier and removed from analysis. Samples were classified as outliers due to PCA plot and tremendous drift of measured values from other samples.

## Liquid chromatography–mass spectrometry analysis of cellular pool sizes of amino acids and polyamines

For amino acid analysis, the benzoylchlorid derivatization method (*Wong et al., 2016*) was used. In brief, 20 µl of the polar phase of each sample, were mixed with 10 µl of 100 mM sodium carbonate (Sigma-Aldrich) followed by the addition of 10 µl 2% benzoylchloride (Sigma-Aldrich) in acetonitrile (VWR). Samples were analyzed using an Acquity iClass UPLC (Waters) connected to a Q-Exactive HF (Thermo Fisher Scientific). For analysis, 1 µl of the derivatized sample was injected onto a 100 mm × 1.0 mm HSS T3 column, packed with 1.8 µm particles (Waters). The flow rate was 100 µl/min and the buffer system consisted of buffer A (10 mM ammonium formate, 0.15% formic acid in water) and buffer B (acetonitrile). The gradient was: 0% B at 0 min; 0–15% B 0–0.1 min; 15–17% B 0.1–0.5 min; 17–55% B 0.5–14 min; 55–70% B 14–14.5 min; 70–100% B 14.5–18 min; 100% B 18–19 min; 100–0% B 19–19.1 min, 19.1–28 min 0% B. The mass spectrometer was operating in positive ionization mode monitoring and the mass range was set to *m/z* 50–750. The heated ESI source settings of the mass spectrometer were: Spray voltage 3.5 kV, capillary temperature 275°C, sheath gas flow 40 AU and aux

gas flow 20 AU at a temperature of 300°C. The S-lens was set to 60 AU. Data analysis was performed using the TraceFinder software (Version 4.1, Thermo Fisher Scientific). Identity of each compound was validated by authentic reference compounds, which were injected and analyzed independently. Extracted ion chromatograms were extracted as $[M+H]^+$ ions with a mass accuracy (<5 ppm). Areas of the cellular pool sizes of the analyzed amines were normalized to their corresponding $^{13}C^{15}N$ internal standard or, if no corresponding $^{13}C^{15}N$ compound was present, they were normalized to the $^{13}C^{15}N$ leucine. Following the normalization to the internal standard the values were normalized to the protein content of the analyzed sample.

## Measurement of RNA stability

To measure mRNA stability, Click-iT Nascent RNA Capture Kit (Invitrogen) was used as described before (*Schatton et al., 2017*). Briefly, 500,000 cells were seeded on 3.5 cm dishes. The following day, endogenous RNA was labeled with 0.2 mM 5-EU for 24 hr and collected either immediately (0-hr time point) or after 8-hr incubation with media w/o EU with Trizol reagent (Invitrogen). Total RNA was isolated according to the instructions of the Trizol reagent manual and 3 μg RNA were biotinylated with biotin-azide using Click-iT chemistry reaction. Afterwards, RNA was precipitated with glycogen, 7.5 M ammonium acetate and 100% ice cold ethanol for 16 hr and centrifugation at 13,000 × *g* for 20 min at 4°C. RNA was washed twice with 70% ethanol, air-dried and resuspended in distilled H2O. 1.5 μg RNA were incubated with prewashed Dynabeads MyOne Streptavidin T1 beads (Invitrogen) for 30 min in presence of RNaseOUT (Invitrogen). Afterwards beads were washed several times and RNA bound on beads was retrotranscribed with SuperScript VILO cDNA synthesis kit (Invitrogen) according to the manual. cDNA was stored at −20°C until qRT-PCR was performed. mRNA half-lives were calculated as described before (*Schatton et al., 2017*).

## RNA isolation, cDNA synthesis, DNA isolation, and quantitative real-time PCR

RNA was isolated with Trizol reagent (Invitrogen) according to the manual. 2 μg of total RNA were retro-transcribed using the SuperScript First-Strand Synthesis System (Invitrogen) with random hexamer primers according to the instructions of the manual. To isolate genomic DNA, cells were lysed with digestion buffer [100 mM NaCl; 10 mM Tris–HCl, pH 8; 25 mM EDTA, pH 8; 0.5% SDS supplemented freshly with 0.1 mg/ml proteinase K (Roche)] at 55°C for 16 hr. DNA was isolated with standard phenol/chloroform purification followed by ethanol precipitation. For mtDNA quantification, 20 ng genomic DNA were used per reaction. SYBR green master mix (Applied Biosystems) was used for quantitative real-time PCR using either StepOne Plus Real-Time PCR system or Quant Studio 12K Flex Real-Time PCR Sytem thermocycler (Applied Biosystems). For each reaction, technical duplicates and at least three biological replicates per experiment were performed. *GAPDH* or *RPL13* were used for normalization and fold enrichment was calculated with the formula: $2^{(-\Delta\Delta Ct)}$. The following primers were used for amplification: *SPAG5* forward: 5'-CATCTCACAGTGGGATAACTAATAAAC-3'; *SPAG5* reverse: 5'-CAGGGATAGGTGAAGCAAGGATA-3'; *GAPDH* forward: 5'-AATCCCATCACCATCTTCCA-3'; *GAPDH* reverse: 5'-TGGACTCCACGACGTACTCA-3'; *RPL13* forward: 5'-CGGACCGTGCGAGGTAT-3'; *RPL13* reverse: 5'-CACCATCCGCTTTTTCTTGTC-3'; *MT-TL1* forward: 5'-CACCCAAGAACAGGGTTTGT-3'; *MT-TL1* reverse: 5'-TGGCCATGGGTATGTTGTTA-3'; *B2M* forward: 5'-TGCTGTCTCCATGTTTGATGTATCT-3'; *B2M* reverse: 5'-TCTCTGCTCCCCACCTCTAAGT-3'.

## Immunofluorescence and transmission electron microscopy

Cells were seeded in an appropriate amount on coverslips the day before. Next day, cells were washed twice with 1× PBS and fixed for 15 min with 4% PFA/PBS (pH7.4) at RT. Afterwards, cells were permeabilized with 0.2% Triton X-100 in 1× PBS for 10 min and blocked in 10% pig or goat serum for 10 min. Then coverslips were stained with primary antibodies diluted in 1% pig or goat serum in 1× PBS for 2 hr at RT or for 16 hr at 4°C. The following primary antibodies were used: rabbit polyclonal anti-CLUH antibody (1:1000, #NB100-93306, Novus Biologicals), mouse monoclonal anti-FLAG antibody (1:1000, #F3165, Sigma-Aldrich), mouse monoclonal anti-PCM1 antibody (1:200, #sc-398365, Santa Cruz Biotechnologies), mouse monoclonal anti-TOMM20 antibody (1:1000, #sc-17764, Santa Cruz Biotechnologies), mouse monoclonal anti-CEP43 antibody (1:500, #WH0011116M1, Sigma-Aldrich), and rabbit polyclonal anti-FLAG antibody (1:1000, #F7425, Sigma-Aldrich). Afterwards, coverslips

were washed three times in 1× PBS for 5 min and incubated with secondary antibodies [donkey anti-rabbit Alexa594 (1:1000, #A21207, Invitrogen), goat anti-mouse Alexa488 (1:1000, #11029, Invitrogen), goat anti-mouse Alexa 594 (1:1000, #A11005, Invitrogen), and goat anti-rabbit Alexa488 (1:1000, #A11034, Invitrogen)] diluted in 1× PBS including 1% pig or goat serum for 1 hr at RT. Then coverslips were washed 3× in 1× PBS (the first washing including DAPI DNA dye) and mounted with Fluorsave reagent (Calbiochem). Transmission electron microscopy on cells was performed as previously described (*Gao et al., 2014*).

## Microscopy and image analysis

Immunofluorescence images were acquired with a spinning-disk confocal microscope (UltraVIEW VoX, PerkinElmer) using a ×60 objective. Images represent a single plane and were deconvoluted using ImageJ (NIH) and brightness was adjusted equally in the entire images. Specificity of the anti-CLUH antibody has been proven before (*Pla-Martín et al., 2020*). For mitochondrial morphology assessment, a half-automated macro for ImageJ (Mitomorph) has been employed (*Yim et al., 2020*). At least 66 (basal) or 44 (galactose) cells per genotype per experiment were analyzed. Micrographs were acquired on a Jeol Jem2100Plus electron microscope operating at a voltage of 120 V using a GATAN OneView camera.

## PI staining and flow cytometry

Cells were synchronized as described before (*Figure 7C*), collected by trypsinization and pelleting, washed once with 1× PBS, fixed in ice-cold 70% ethanol in 1× PBS and stored at −20°C for at least 16 hr. Fixed cells were pelleted at 2000 rpm for 10 min, washed once with 1× PBS and pelleted again at 2000 rpm for 5 min. Pellets were resuspended in 500 µl 1× PBS containing 0.25% Triton X-100, 100 µg/ml RNase A (#1007885, Qiagen) and 50 µg/ml PI (#P4864, Sigma-Aldrich) and incubated at least 30 min at RT before measured by flow cytometry. BD LSR Fortessa (BD Biosciences) was used with BD FACS Diva software at low flow rate using PE laser (561 nm excitation; 586/15 nm detection). At least 15,000 events were measured per sample. Flowing Software version 2.5.1 (developed by Perttu Terho, Turku Centre for Biotechnology, University of Turku, Finland) was used for analysis.

## Statistics

Sample size has been determined by previous experience with similar analysis. Replicates are always biological independent experiments. Data are shown as mean ± standard error of the mean or ± standard deviation as indicated in respective figure legends. To compare two groups, paired or unpaired Student's *t*-test was performed as indicated in the figure legends. To compare multiple groups, one-way analysis of variance (ANOVA) with post hoc Tukey's multiple comparison test was performed. To compare datasets including two variances, two-way ANOVA with post hoc Tukey's multiple comparison test was employed. Statistical significance was calculated using GraphPad Prism software. A p value <0.05 was considered as significant. Statistical methods for proteomics analysis are described in the corresponding method section.

## Acknowledgements

We are grateful to Guy Lenaers for providing *CLUH* knockout HeLa cells, and to Iain Cheeseman for sharing the *SPAG5* and *KNSTRN* ind-KO cell lines. We thank the CECAD imaging and proteomics facilities for excellent technical assistance, and Hisham Bazzi and members of the Rugarli laboratory for constructive discussions. This work was funded by the Deutsche Forschungsgemeinschaft (Project numbers 269925409 and 411422114-GRK 2550) to E.I.R.

## Additional information

### Funding

| Funder | Grant reference number | Author |
|---|---|---|
| Deutsche Forschungsgemeinschaft | 269925409 | Elena I Rugarli |
| Deutsche Forschungsgemeinschaft | 411422114-GRK 2550 | Elena I Rugarli |
| Max Planck Institute for Biology of Ageing | open access funding | Thomas Langer |

The funders had no role in study design, data collection, and interpretation, or the decision to submit the work for publication.

### Author contributions

Désirée Schatton, Conceptualization, Formal analysis, Investigation, Validation, Visualization, Writing – original draft, Writing – review and editing; Giada Di Pietro, Karolina Szczepanowska, Stefan Müller, Patrick Giavalisco, Formal analysis, Investigation, Writing – review and editing; Matteo Veronese, Investigation, Writing – review and editing; Marie-Charlotte Marx, Kristina Braunöhler, Esther Barth, Investigation; Thomas Langer, Conceptualization, Writing – review and editing; Aleksandra Trifunovic, Conceptualization; Elena I Rugarli, Conceptualization, Funding acquisition, Project administration, Supervision, Validation, Writing – original draft, Writing – review and editing

### Author ORCIDs

Patrick Giavalisco http://orcid.org/0000-0002-4636-1827
Aleksandra Trifunovic http://orcid.org/0000-0002-5472-3517
Elena I Rugarli http://orcid.org/0000-0002-5782-1067

### Decision letter and Author response

Decision letter https://doi.org/10.7554/eLife.74552.sa1
Author response https://doi.org/10.7554/eLife.74552.sa2

## Additional files

### Supplementary files

- Supplementary file 1. Interactors of endogenous human CLUH in HeLa cells after SILAC labeling.
- Supplementary file 2. Interactors of endogenous human CLUH in G2-enriched HeLa cells.
- Supplementary file 3. Proteomics data of *SPAG5* and *KNSTRN* ind-KO HeLa cells.
- Supplementary file 4. IC-TQ_Transitions compound list.
- Transparent reporting form

### Data availability

Source data files have been provided for Figures 1, 2, 3, 4, 5, 6 and 7. The mass spectrometry proteomic data have been deposited to the ProteomeXchange Consortium via the PRIDE partner repository with the dataset identifiers: PXD029142, PXD029145, PXD029156.

The following datasets were generated:

| Author(s) | Year | Dataset title | Dataset URL | Database and Identifier |
|---|---|---|---|---|
| Rugarli EI | 2022 | CLUH controls astrin-1 expression to couple mitochondrial metabolism to cell cycle progression | https://www.ebi.ac.uk/pride/archive/projects/PXD029142 | PRIDE, PXD029142 |

*Continued*

| Author(s) | Year | Dataset title | Dataset URL | Database and Identifier |
|-----------|------|---------------|-------------|------------------------|
| Rugarli EI | 2022 | CLUH controls astrin-1 expression to couple mitochondrial metabolism to cell cycle progression | https://www.ebi.ac.uk/pride/archive/projects/PXD029145 | PRIDE, PXD029145 |
| Rugarli EI | 2022 | CLUH controls astrin-1 expression to couple mitochondrial metabolism to cell cycle progression | https://www.ebi.ac.uk/pride/archive/projects/PXD029156 | PRIDE, PXD029156 |

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

# Appendix 1

## Appendix 1—key resources table

| Reagent type (species) or resource | Designation | Source or reference | Identifiers | Additional information |
|---|---|---|---|---|
| Gene (*Homo sapiens*) | *CLUH* | NCBI | NM_001366661.1 | |
| Gene (*Homo sapiens*) | *SPAG5* | NCBI | NM_006461.3 | |
| Gene (*Homo sapiens*) | *KNSTRN* | NCBI | NM_033286.4 | |
| Cell line (*Homo sapiens*) | *CLUH* KO HeLa cells | **Wakim et al., 2017** | | Exon 4 has been deleted with CRISPR-Cas9 technique |
| Cell line (*Homo sapiens*) | Doxycyline inducible WT HeLa cells | **McKinley and Cheeseman, 2017** | | Cas9 expression is induced by doxycycline treatment; constitutive sgRNA expression |
| Cell line (*Homo sapiens*) | Inducible *SPAG5* KO HeLa cells | **McKinley and Cheeseman, 2017**; **Kern et al., 2017** | | Cas9 expression is induced by doxycycline treatment; constitutive sgRNA expression targeting *SPAG5* |
| Cell line (*Homo sapiens*) | Inducible *KNSTRN* KO HeLa cells | **McKinley and Cheeseman, 2017**; **Kern et al., 2017** | | Cas9 expression is induced by doxycycline treatment; constitutive sgRNA expression targeting *KNSTRN* |
| Cell line (*Homo sapiens*) | HEK293T cells expressing FLAG-kinastrin | This paper | | Doxycyline inducible |
| Cell line (*Homo sapiens*) | *SPAG5* KO HeLa cells | This paper | | Stable KO cells; derived by monoclonal selection of inducible *SPAG5* KO HeLa cells |
| Cell line (*Homo sapiens*) | *KNSTRN* KO HeLa cells | This paper | | Stable KO cells; derived by monoclonal selection of inducible *KNSTRN* KO HeLa cells |
| Cell line (*Mus musculus*) | *Cluh* KO MEFs | **Gao et al., 2014** | | Exon 10 has been deleted with Cre/LoxP system |
| Recombinant DNA reagent | p3xFLAG-CMV-14 (plasmid) | Sigma-Aldrich | | Backbone of all SPAG5 constructs; used as empty vector control |
| Transfected construct (human) | ATG1-SPAG5 | This paper | | Allows the expression of a FLAG-tagged human SPAG5 from ATG1 |
| Transfected construct (human) | Δ151-SPAG5 | This paper | | Allows the expression of a FLAG-tagged human SPAG5 lacking the first 151 amino acids |
| Transfected construct (human) | 5UTR-SPAG5 | This paper | | Allows the expression of a FLAG-tagged human SPAG5 including its 5′UTR; used to overexpress astrin-1 and 2 together |
| Transfected construct (human) | uATG$^{GGG}$-SPAG5 | This paper | | Allows the expression of a FLAG-tagged human SPAG5 including its 5′UTR with uATG mutated to GGG |
| Transfected construct (human) | ATG1$^{GGG}$-SPAG5 | This paper | | Allows the expression of a FLAG-tagged human SPAG5 including its 5′UTR with ATG1 mutated to GGG; used to overexpress only astrin-2 |
| Transfected construct (human) | ATG3$^{GGG}$-SPAG5 | This paper | | Allows the expression of a FLAG-tagged human SPAG5 including its 5′UTR with ATG3 mutated to GGG |
| Transfected construct (human) | ATG4$^{GGG}$-SPAG5 | This paper | | Allows the expression of a FLAG-tagged human SPAG5 including its 5′UTR with ATG4 mutated to GGG |
| Transfected construct (human) | ATG5$^{GGG}$-SPAG5 | This paper | | Allows the expression of a FLAG-tagged human SPAG5 including its 5′UTR with ATG5 mutated to GGG |
| Transfected construct (human) | ATG6$^{GGG}$-SPAG5 | This paper | | Allows the expression of a FLAG-tagged human SPAG5 including its 5′UTR with ATG6 mutated to GGG |
| Transfected construct (human) | ATG3 + 5$^{GGG}$-SPAG5 | This paper | | Allows the expression of a FLAG-tagged human SPAG5 including its 5′UTR with ATG3 and 5 mutated to GGG |
| Transfected construct (human) | ATG3 + 4 + 5$^{GGG}$-SPAG5 | This paper | | Allows the expression of a FLAG-tagged human SPAG5 including its 5′UTR with ATG3, 4, and 5 mutated to GGG; used to overexpress only astrin-1 |
| Transfected construct (human) | ATG3 + 4 + 5 + 6$^{GGG}$-SPAG5 | This paper | | Allows the expression of a FLAG-tagged human SPAG5 including its 5′UTR with ATG3, 4, 5, and 6 mutated to GGG |
| Transfected construct (human) | ctrl siRNA | Invitrogen | #12935115 | Transfected construct (human) |

*Appendix 1 Continued on next page*

*Appendix 1 Continued*

| Reagent type (species) or resource | Designation | Source or reference | Identifiers | Additional information |
|---|---|---|---|---|
| Transfected construct (human) | siRNA targeting *CLUH* | Invitrogen | CLUHHSS118510 | Transfected construct (human) |
| Transfected construct (human) | siRNA targeting *CLUH* | Invitrogen | CLUHHSS118511 | Transfected construct (human) |
| Transfected construct (human) | siRNA targeting *CLUH* | Invitrogen | CLUHHSS177299 | Transfected construct (human) |
| Antibody | Anti-CLUH (rabbit polyclonal) | Novus Biologicals | #NB100-93305 | WB (1:1000) IP (0.5 µg /300–500 µg protein) |
| Antibody | Anti-CLUH (rabbit polyclonal) | Novus Biologicals | #NB100-93306 | IF(1:1000), WB (1:1000) IP (0.5 µg /300–500 µg protein) |
| Antibody | Anti-RPS6 (rabbit polyclonal) | Novus Biologicals | #NB100-1595 | WB (1:1000) |
| Antibody | Anti-RPL7 (rabbit polyclonal) | Novus Biologicals | #NB100-2269 | WB (1:1000) |
| Antibody | Anti-astrin (rabbit polyclonal) | ProteinTech | #14726-1-AP | WB (1:1000) IP (0.5 µg /300–500 µg protein) |
| Antibody | Anti-FLAG (rabbit polyclonal) | Sigma-Aldrich | #F7425 | IF (1:1000), WB (1:1000) IP (0.5 µg /300–500 µg protein) |
| Antibody | Anti-FLAG (mouse monoclonal) | Sigma-Aldrich | #F3165 | IF (1:1000), WB (1:1000) |
| Antibody | Pan-actin (mouse monoclonal) | EMD Millipore | #MAB1501 | WB (1:2000) |
| Antibody | Anti-GAPDH (mouse monoclonal) | EMD Millipore | #MAB374 | WB (1:2000) |
| Antibody | Anti-CLUH (rabbit polyclonal) | Aviva | #ARP70642_P050 | WB (1:1000) |
| Antibody | Anti-kinastrin (rabbit polyclonal) | Abcam | #ab122769 | WB (1:1000) |
| Antibody | pH3-Ser10 (rabbit polyclonal) | Abcam | #ab5176 | WB (1:1000) |
| Antibody | Anti-SDHA (mouse monoclonal) | Molecular probes | #459,200 | WB (1:1000) |
| Antibody | Anti-NDUFA9 (mouse monoclonal) | Molecular probes | #459,100 | WB (1:1000) |
| Antibody | Anti-UQCRC1 (mouse monoclonal) | Molecular probes | #459,140 | WB (1:1000) |
| Antibody | pRB1-Ser807/811 (rabbit polyclonal) | Cell Signaling | #9308 | WB (1:1000) |
| Antibody | pCDK1-Tyr15 (rabbit monoclonal) | Cell Signaling | #4539 | WB (1:1000) |
| Antibody | Anti-cyclin D3 (mouse monoclonal) | Cell Signaling | #2936 | WB (1:1000) |

*Appendix 1 Continued*

| Reagent type (species) or resource | Designation | Source or reference | Identifiers | Additional information |
|---|---|---|---|---|
| Antibody | pRPS6-Ser235/236 (rabbit polyclonal) | Cell Signaling | #2211 | WB (1:1000) |
| Antibody | Anti-CDK1 (mouse monoclonal) | Santa Cruz Biotechnologies | #sc-54 | WB (1:1000) |
| Antibody | Anti-kinastrin (rabbit polyclonal) | Sigma-Aldrich | #SAB1103031 | WB (1:1000) IP (0.5 µg /300–500 µg protein) |
| AntibodyA | Anti-astrin (rabbit polyclonal) | Novus Biologicals | #NB100-74638 | IP (0.5 µg /300–500 µg protein) |
| Antibody | Anti-AFG3L1 (rabbit polyclonal) | *Koppen et al., 2007* | N/A | IP (0.5 µg /300–500 µg protein) |
| Antibody | Anti-TOMM20 (mouse monoclonal) | Santa Cruz Biotechnologies | #sc-17764 | IF (1:1000) |
| Antibody | Anti-PCM1 (mouse monoclonal) | Santa Cruz Biotechnologies | #sc-398365 | IF (1:200) |
| Antibody | Anti-CEP43 (mouse monoclonal) | Sigma-Aldrich | #WH0011116M1 | IF (1:500) |
| Antibody | Anti-rabbit Alexa594 (donkey polyclonal) | Invitrogen | #A21207 | IF (1:1000) |
| Antibody | Anti-mouse Alexa488 (goat polyclonal) | Invitrogen | #A11029 | IF (1:1000) |
| Antibody | Anti-mouse Alexa594 (goat polyclonal) | Invitrogen | #A11005 | IF (1:1000) |
| Antibody | Anti-rabbit Alexa488 (goat polyclonal) | Invitrogen | #A11034 | IF (1:1000) |
| Sequence-based reagent | SPAG5 forward | This paper | PCR primers | 5'-CATCTCACAGTGGGATAACTAATAAAC-3' |
| Sequence-based reagent | SPAG5 reverse | This paper | PCR primers | 5'-CAGGGATAGGTGAAGCAAGGATA-3' |
| Sequence-based reagent | GAPDH forward | This paper | PCR primers | 5'-AATCCCATCACCATCTTCCA-3' |
| Sequence-based reagent | GAPDH reverse | This paper | PCR primers | 5'-TGGACTCCACGACGTACTCA-3' |
| Sequence-based reagent | RPL13 forward | This paper | PCR primers | 5'-CGGACCGTGCGAGGTAT-3' |
| Sequence-based reagent | RPL13 reverse | This paper | PCR primers | 5'-CACCATCCGCTTTTTCTTGTC-3' |
| Sequence-based reagent | MT-TL1 forward | *Rooney et al., 2015* | PCR primers | 5'-CACCCAAGAACAGGGTTTGT-3' |
| Sequence-based reagent | MT-TL1 reverse | *Rooney et al., 2015* | PCR primers | 5'-TGGCCATGGGTATGTTGTTA-3' |
| Sequence-based reagent | B2M forward | *Rooney et al., 2015* | PCR primers | 5'-TGCTGTCTCCATGTTTGATGTATCT-3' |
| Sequence-based reagent | B2M reverse | *Rooney et al., 2015* | PCR primers | 5'-TCTCTGCTCCCCACCTCTAAGT-3' |

*Appendix 1 Continued on next page*

*Appendix 1 Continued*

| Reagent type (species) or resource | Designation | Source or reference | Identifiers | Additional information |
|---|---|---|---|---|
| Commercial assay or kit | Click-it Nascent RNA Capture Kit | Invitrogen | C10365 | |
| Commercial assay or kit | SuperScript VILO cDNA synthesis kit | Invitrogen | 11754250 | Used in combination with Click-it Nascent RNA Capture Kit |
| Commercial assay or kit | SuperScript First-Strand Synthesis System | Invitrogen | 11904018 | |
| Chemical compound, drug | Doxycycline | Sigma-Aldrich | #D5207 | Final conc.: 1 µg/ml |
| Chemical compound, drug | L-Arginine-HCl | Silantes | #201604102 | Final conc.: 28 µg/ml |
| Chemical compound, drug | L-Lysine-HCl | Silantes | #211604102 | Final conc.: 73 µg/ml |
| Chemical compound, drug | Cycloheximide | Sigma-Aldrich | #C4859 | Final conc.: 100 µg/ml |
| Chemical compound, drug | MG132 | Sigma-Aldrich | #M8699 | Final conc.: 20 µM |
| Chemical compound, drug | Thymidine | Sigma-Aldrich | #T1895 | Final conc.: 2 mM |
| Chemical compound, drug | Nocodazole | Sigma-Aldrich | #M1404 | Final conc.: 100 ng/µl |
| Chemical compound, drug | Benzonase HC nuclease | Sigma-Aldrich | #71206 | Final amount per samples: 25 U |
| Chemical compound, drug | Propidium iodide | Sigma-Aldrich | #P4864 | Final conc.: 50 µg/ml |

