## [Editor Report]

The work is of interest to cell biologists studying metabolism and its regulation during the cell cycle. It reveals how CLUH, a protein involved in mitochondrial function regulation and metabolism, regulates the levels of astrin, a protein functionally involved in cell division and mTOR regulator, integrating metabolism and cell cycle.

---

## [Decision Letter]

**Decision letter after peer review:**

Thank you for submitting your article "CLUH controls astrin-1 expression to couple mitochondrial metabolism to cell cycle progression" for consideration by *eLife*. Your article has been reviewed by 2 peer reviewers, including Agnieszka Chacinska as the Reviewing Editor and Reviewer #1, and the evaluation has been overseen by Erica Golemis as the Senior Editor.

Essential revisions:

The authors should perform additional experiments in accord with the remarks appended in the individual reviews, with a special focus on the issue of localization. They should also clarify noted statements and interpretations concerning the effect on metabolism.

*Reviewer #1 (Recommendations for the authors):*

I highly value the work presented in the manuscript, the quality of data, their presentation and interpretation. Several burning questions arise and certainly they cannot be answered in one study.

Having understood that uncovering spatial regulation requires additional deep investigation efforts beyond the scope of this manuscript, it is suggested to limit some possible scenarios and address some (limited) aspects of the cellular distribution of astrin and Cluh. The fact that Cluh interacts with both mRNA and astrin-1 may suggest the role in some kind of translational hot-spots, acting rather far away from chromosomes and during interphase. The further interactions of Cluh with microbular and centrosome proteins do not exclude such a scenario. On the other hand the localization of Cluh to centrosomes, and its dependence on astrin-1 is not demonstrated, as the granular structures of Cluh in the absence of astrin remains enigmatic. Having some of these basics clarified would allow to have a more focused discussion concerning this aspect.

The authors are also asked to make sure that critical findings on conditional knockouts are controlled for specificity.

*Reviewer #2 (Recommendations for the authors):*

1) The authors beautifully demonstrate that several downstream ATGs are responsible for the formation of the smaller isoform Astrin-2. This suggests that they are made from the same mRNA. As they show later on that loss of CLUH destabilizes the SPAG5 mRNA, why would only be the synthesis of the longer isoform be affected? The trend is observed in the figure but not mentioned in the text. If there is a true difference, how would CLUH only affect translation of one astrin isoform and not the other? Is this connected to the region of the mRNA that is bound by CLUH?

2) Astrin overexpression beautifully regulates CLUH localization. I would really appreciate some blow ups of the images or arrows pointing to the dots here to make it more obvious.

3) Could expression of astrin1 vs 2 rescue CLUH localization in SPAG5 KO cells? If yes this would be an important tool to untangle the impact of CLUH mislocalization from loss of astrin.

4) Metabolomics of starved CLUH vs starved Astrin KO cells are polar opposites, which the authors suggest could be due to CLUH also regulating the expression of TCA enzymes etc. via its RNA-binding function. This seems reasonable, but is confounded by the fact that loss of astrin will directly impact mTOR via Raptor, but also relocalize CLUH. As mentioned above, re-expression of astrin2, which judging by Thedieck et al. 2013 still binds Raptor, in SPAG5 KO may help to untangle the direct impact of astrin on mTOR and the indirect one via CLUH, which the authors have previously proposed to regulate mTOR by sequestering it into G3BP1-containing granules (Pla-Martin et al. 2020).

5) Judging by Suppl Figure 6 the combination of CLUH and astrin loss on mTOR signaling is additive, which supports the hypothesis that CLUH affects mTOR via a pathway separate from astrin, probably via G3BP1. How does loss of G3BP1 impact mTOR activation in SPAG5 KO?

6) The authors claim a "different temporal profile for CDK1 on Tyr15". This needs to be more precisely defined; to me, the temporal pattern looks similar but the magnitude is higher in CLUH KO cells.

---

## [Author Response]

Essential revisions:The authors should perform additional experiments in accord with the remarks appended in the individual reviews, with a special focus on the issue of localization. They should also clarify noted statements and interpretations concerning the effect on metabolism.

We are grateful for the Reviewers’ comments and have performed additional experiments to shed light on the localization of CLUH and its dependence on astrin. In addition, we have revised the text to better reflect the results and remove interpretations that are at present only speculative. A detailed description of all changes is listed below in the point-by-point response to the Reviewers.

Reviewer #1 (Recommendations for the authors):I highly value the work presented in the manuscript, the quality of data, their presentation and interpretation. Several burning questions arise and certainly they cannot be answered in one study.Having understood that uncovering spatial regulation requires additional deep investigation efforts beyond the scope of this manuscript, it is suggested to limit some possible scenarios and address some (limited) aspects of the cellular distribution of astrin and Cluh. The fact that Cluh interacts with both mRNA and astrin-1 may suggest the role in some kind of translational hot-spots, acting rather far away from chromosomes and during interphase. The further interactions of Cluh with microbular and centrosome proteins do not exclude such a scenario. On the other hand the localization of Cluh to centrosomes, and its dependence on astrin-1 is not demonstrated, as the granular structures of Cluh in the absence of astrin remains enigmatic. Having some of these basics clarified would allow to have a more focused discussion concerning this aspect.The authors are also asked to make sure that critical findings on conditional knockouts are controlled for specificity.

We thank the Reviewer for the positive comments and the insightful suggestions. We have now performed additional experiments to shed light on a possible localization of CLUH at the centrosome and how it may depend on astrin-1. Mild overexpression of astrin-1 leads to the formation of a punctate perinuclear structure that resembles the centrosome. Using markers for the centrosomes (CEP43) and the pericentriolar material (PCM1), we confirmed that astrin-1 localizes to these structures, as previously demonstrated. Cells overexpressing astrin-1 can also form additional aggregates that are not coincident with the centrosome. However, these experiments should be taken with caution, since they are performed upon overexpression. More experiments will be required to understand the exact nature of these additional structures. We now show examples for both types of occurrence in the revised Figure 3D.

We have also performed experiments to test if endogenous CLUH localizes at the centrosome. Analysis of thousands of cells in unsynchronized cultures did not reveal staining of endogenous CLUH at the centrosome (Figure 3E). While these experiments cannot exclude the presence of CLUH at the centrosome during a particular phase of the cell cycle, we can rule out the relevance of such a localization for the main function of CLUH during interphase.

To further investigate how lack of astrin impacts on CLUH subcellular localization, and to control for specificity of experiments in the *SPAG5* KO clones, we performed rescue experiments by transiently re-expressing different astrin isoforms in *SPAG5* KO cells (see also response to Reviewer # 2, point 3). However, we could not rescue the abnormal localization of CLUH at the periphery of the cells by expressing astrin-1, astrin-2 or both. Surprisingly, these proteins were also recruited at focal adhesions together with CLUH. It is possible that this abnormal CLUH localization is a clone-specific feature, or that it reflects cell cycle defects observed in these cells, or a metabolic phenotype that cannot be rescued by transient overexpression. Moreover, to detect this localization of CLUH at focal adhesion it is important to culture the cells at high confluence. To fully understand the significance of these CLUH-positive structures, more experiments and the generation of rescue clones with endogenous levels of astrin re-expression are required. We believe that these experiments are outside the scope of the current manuscript. We have therefore decided to remove these data from the manuscript, as they are too preliminary and their interpretation is speculative. We have amended the discussion accordingly. We think that omitting these data does not change the main message of our manuscript that focuses on revealing how CLUH regulates astrin-1 levels and integrates mitochondrial metabolism with the cell cycle.

We show in Author response image 1 the rescue experiment results to the Reviewers.

**Author response image 1. sa2fig1:** Confocal immunofluorescence pictures of HeLa cells overexpressing for 24 h FLAG-tagged astrin-1 and astrin-2, astrin-1 or astrin-2 alone stained for FLAG (red) and CLUH (green). DAPI was used to stain nuclei (blue). Small boxes in corners show a 4x magnified area of boxed regions. Scale bar, 8 µm.

Reviewer #2 (Recommendations for the authors):1) The authors beautifully demonstrate that several downstream ATGs are responsible for the formation of the smaller isoform Astrin-2. This suggests that they are made from the same mRNA. As they show later on that loss of CLUH destabilizes the SPAG5 mRNA, why would only be the synthesis of the longer isoform be affected? The trend is observed in the figure but not mentioned in the text. If there is a true difference, how would CLUH only affect translation of one astrin isoform and not the other? Is this connected to the region of the mRNA that is bound by CLUH?

The Reviewer is right in noticing that there is a tendency towards reduction of the synthesis also of astrin-2, as it would be expected if CLUH stabilizes the *SPAG5* mRNA. We have now performed an additional experiment to increase the biological replicates, however we still observed a consistent reduction of the newly synthesized astrin-1 while the synthesis of astrin-2 was affected in some experiments but not in others. One problem in the interpretation of this experiment is that astrin-1 is also immediately stabilized in wild-type cells by binding CLUH. We therefore likely observe here a combination of the effects that lack of CLUH has on translation and on protein stability of astrin-1. For this reason, we have now rephrased in a more cautionary tone our conclusion in the Result section: “We observed that newly synthesized astrin-1 is reduced in absence of CLUH to a greater extent than astrin-2, probably reflecting the decreased stability of the longer isoform (Figure 2H-I).” Moreover, we have added the following sentence to the Discussion: “While we established the role of CLUH binding to prevent astrin-1 degradation, more experiments are required to determine to what extent the synthesis of astrin-2 is also modulated by CLUH.”

2) Astrin overexpression beautifully regulates CLUH localization. I would really appreciate some blow ups of the images or arrows pointing to the dots here to make it more obvious.

These experiments have now been inserted in Figure 1G and enlargements have been provided as requested by the Reviewer.

3) Could expression of astrin1 vs 2 rescue CLUH localization in SPAG5 KO cells? If yes this would be an important tool to untangle the impact of CLUH mislocalization from loss of astrin.

We thank the Reviewer for raising this point. We have performed rescue experiments by transiently re-expressing different astrin isoforms in *SPAG5* KO cells (see also response to Reviewer # 1). However, we could not rescue the abnormal localization of CLUH at the periphery of the cells by expressing astrin-1, astrin-2 or both. Surprisingly, these proteins were also recruited at focal adhesions together with CLUH. It is possible that this abnormal CLUH localization is a clone-specific feature, or that it reflects cell cycle defects observed in these cells, or a metabolic phenotype that cannot be rescued by transient overexpression. Moreover, to detect this localization of CLUH at focal adhesion it is important to culture the cells at high confluence. To fully understand the significance of these CLUH-positive structures, more experiments and the generation of rescue clones with endogenous levels of astrin re-expression are required. We believe that these experiments are outside the scope of the current manuscript. We have therefore decided to remove these data from the manuscript, as they are too preliminary and their critical discussion too speculative. We have amended the discussion accordingly. We think that omitting these data does not change the main message of our manuscript that focuses on revealing how CLUH regulates astrin-1 levels and integrates mitochondrial metabolism with the cell cycle.

We show the rescue experiments results to the Reviewers (see response to Reviewer # 1).

4) Metabolomics of starved CLUH vs starved Astrin KO cells are polar opposites, which the authors suggest could be due to CLUH also regulating the expression of TCA enzymes etc. via its RNA-binding function. This seems reasonable, but is confounded by the fact that loss of astrin will directly impact mTOR via Raptor, but also relocalize CLUH. As mentioned above, re-expression of astrin2, which judging by Thedieck et al. 2013 still binds Raptor, in SPAG5 KO may help to untangle the direct impact of astrin on mTOR and the indirect one via CLUH, which the authors have previously proposed to regulate mTOR by sequestering it into G3BP1-containing granules (Pla-Martin et al. 2020).

The Reviewer is asking if astrin regulates mTORC1 only via its ability to sequester raptor in stress granules, or also by enhancing CLUH function. Answering this question requires the generation of cell lines that express only astrin-2 or astrin-1 at the endogenous level in absence of *SPAG5*. Although we have attempted to generate these cell lines during the time of the revision, we were not successful. Transient experiments are not suited to this purpose, given the high variability of mTORC1 signaling among cells and different experiments. We agree with the Reviewer that this is an important point to investigate in the future, however it is not in the focus of our current manuscript. Our main message here is that CLUH regulates astrin-1 and in this way couples mitochondrial metabolism with the cell cycle. This opens several questions on the role of astrin itself, especially in the context of cancer biology, which are outside the scope of the current manuscript. In addition, it is possible that the mTORC1 hyperactivation in cells lacking *SPAG5* is the result of a more complicated signaling dysregulation than what was reported by Thedieck at al. To unravel this abnormal mTORC1 signaling will require many more experiments and is a story on its own.

5) Judging by Suppl Figure 6 the combination of CLUH and astrin loss on mTOR signaling is additive, which supports the hypothesis that CLUH affects mTOR via a pathway separate from astrin, probably via G3BP1. How does loss of G3BP1 impact mTOR activation in SPAG5 KO?

In Suppl Figure 6, the combination of CLUH and astrin loss shows indeed an additive effect on mTORC1 activation. However, an alternative explanation is that CLUH downregulation further depletes astrin, also from those cells that were not targeted by the inducible CRISPR-Cas9 system. Therefore, any conclusion based on this experiment is premature.

We have tested if *G3BP1* downregulation impacts mTORC1 activation in *SPAG5* KO cells (using total KO cells to avoid the problem mentioned above). These experiments show that depletion of G3BP1 leads to mTORC1 hyperactivation (in agreement with data from Prentzell et al., 2021, PMID: 33497611), and this is not significantly enhanced in *SPAG5* KO cells, at least in four independent experiments (see Author response image 2). However, we think that adding these data to the paper will only trigger additional questions, and we would therefore prefer to leave them outside the manuscript. Dissecting in depth if astrin and CLUH regulate mTORC1 in separate ways and the role of G3BP1 is a long-term project. Although we are interested in further exploring this question, we think that it is not important for the conclusion of the current paper. In fact, the only point that we want to make here is that, despite a similar abnormal signaling of mTORC1 upon starvation, the metabolic profile of *CLUH* and *SPAG5* KO cells is very different. We also want to point out that CLUH is regulating the expression of mitochondrial proteins that are required to fully allow the metabolic reprogramming induced by mTORC1 signaling. We have now added in the Discussion the following sentence: “It remains to be established if mTORC1 hyperactivation observed upon loss of CLUH is entirely dependent on astrin destabilization, or whether different pathways are at play.”

**Author response image 2. sa2fig2:** (A) Western blot for the indicated antibodies in WT and *SPAG5* KO cells downregulated with control of *G3BP1* siRNA in basal and HBSS conditions. (B) Quantification of four biological replicates. One-way ANOVA with post-hoc Tukey’s test. P< 0.01 **.

6) The authors claim a "different temporal profile for CDK1 on Tyr15". This needs to be more precisely defined; to me, the temporal pattern looks similar but the magnitude is higher in CLUH KO cells.

We thank the Reviewer for this comment. We have now performed additional experiments and quantified the temporal pattern of all the antibodies used. Results have been added to Figure 7I. The Reviewer was indeed right and the temporal profile of CDK1-Tyr15 was not changed, while this quantification revealed different profiles of pRB1, cyclin D3, and astrin-1. We have also tested if the levels of these markers changed in unsynchronized cells, and added these new data in the revised Figure 7A.